# A listening advantage for native speech is reflected by attention-related activity in auditory cortex
Meng Liang [1], Johannes Gerwien [2] & Alexander Gutschalk [1] ✉

The listening advantage for native speech is well known, but the neural basis of the effect remains unknown. Here we test the hypothesis that attentional enhancement in auditory cortex is stronger for native speech, using magnetoencephalography. Chinese and German speech stimuli were recorded by a bilingual speaker and combined into a two-stream, cocktail-party scene, with consistent and inconsistent language combinations. A group of native speakers of Chinese and a group of native speakers of German performed a detection task in the cued target stream. Results show that attention enhances negative-going activity in the temporal response function deconvoluted from the speech envelope. This activity is stronger when the target stream is in the native compared to the non-native language, and for inconsistent compared to consistent language stimuli. We interpret the findings to show that the stronger activity for native speech could be related to better top-down prediction of the native speech streams.

Listening to native speech is an effortless task in quiet environments. Some attentional effort is required when there are competing speakers or other additional sound sources, but we can usually segregate a target speech stream from the background[1]. However, competing speech streams can mask part of the information in the target stream. The degree of this masking depends on a multitude of factors. First, there is direct energetic masking in the cochlea, which depends on the spectral overlap of the speech and masker signals[2]. Second, the similarity of the target and masker streams influences how well the target speech can be perceived and understood. For example, it is easier to segregate speech streams with different genders than same-gender speakers and more difficult to segregate two speech segments from the same speaker[3]. These aspects of the masking are closely related to basic auditory scene analysis mechanisms[4], which determine how well two auditory streams can be segregated in the auditory system, relying on cues such as frequency difference[5], pitch[6], spatial cues[7], or timbre[8,9]. Neural correlates for these basic stream segregation mechanisms have been found in the auditory cortex[10–14] and to some degree in the brainstem[15,16]. Beyond these basic parameters, linguistic aspects play a role in the interference between two speech streams, which can be observed in the comparison of native and non-native speech[17]. For example, informational masking decreases performance more strongly when attending to non-native speech[18], and native-speech maskers produce stronger interference with the target than non-native-speech maskers[19]. The disadvantage for non-native target speech depends on the linguistic complexity of the context,

with stronger effects for complex sentences (with syntax and meaning) than simple contexts (word lists)[20]. In contrast, the pure energetic masking caused by noise does not seem to affect native and non-native speakers disproportionally on a lower, phonetic level[21].

Electroencephalography (EEG), magnetoencephalography (MEG), and intracranial EEG (iEEG) studies have shown that activity in the auditory cortex time-locked to the speech envelope is enhanced for the attended speaker in two-speaker scenes[22–25]. In EEG and MEG, this effect is observed for a negative-going response with a latency of 100 ms and longer. In the context of speech streams, this negative-going wave has mostly been referred to as N1/N100[22], whereas studies investigating attention to tone streams have variably referred to it as N1[26], processing negativity[27,28], or neutrally to the negative difference wave (Nd)[29,30]. Because in our experience, the negative-going enhancement for attended speech streams overlaps the latency range of the N1 and P2 components of non-attended streams[31], we here adopt the latter nomenclature and refer to the Nd response to characterize the enhancement observed for attended speech streams. This attentional modulation of the target-speech-evoked signal is highly context-dependent; the comparison of a meaningful speech and a non-meaningful, random speech masker revealed stronger target-speech-related activity with the non-meaningful masker in the left hemisphere[32]. In a situation where two streams spoken by the same speaker were used, attentional enhancement was not observed for the cued target speech, unless the whole stream was primed beforehand by prior presentation of a single speech stream[33].

[1]Department of Neurology, University of Heidelberg, Im Neuenheimer Feld 400, 69120 Heidelberg, Germany. [2]Institute of German as a Foreign Language Philology, University of Heidelberg, Plöck 55, 69117 Heidelberg, Germany. ✉e-mail: Alexander.Gutschalk@med.uni-heidelberg.de

**Fig. 1 | Behavioral performance for the detection of the target name, for native Chinese ($n = 17$) and German ($n = 17$) listeners (mean ± standard error; circles indicate single-subject data). a, b** Hit rates for the target name in the cued target stream. **c, d** False alarm rates, i.e., erroneous detections of the target word in the masker stream. **e, f** showed the resulting detectability ($d'$). Stimulus language combinations are coded by color (red/C–C: consistent, Chinese target; black/G–G: consistent, German target; green/C–G: inconsistent, Chinese target; blue/G–C: inconsistent German target). Primed conditions are indicated by hatched, and unprimed conditions by solid bars.

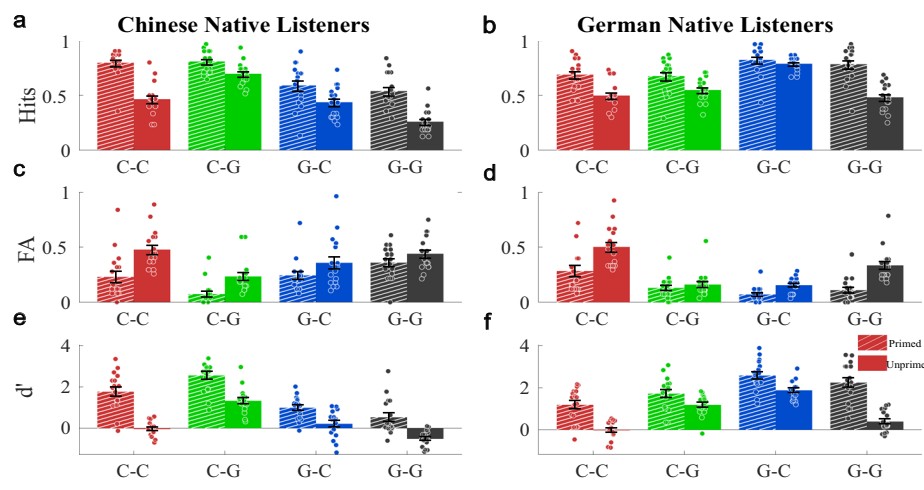

## Table 1 | Statistical analysis of behavioral target detectability ($d'$)

| **Unprimed, two streams** | | | |
|---|---|---|---|
| Test | $F (1,32)$ | $p$ | Partial $\eta^2$ |
| Target language | 1.990 | 0.168 | 0.059 |
| Target language * group | 64.729 | <0.001 | 0.669 |
| T–M consistency | 250.377 | <0.001 | 0.887 |
| T–M consistency * group | 4.030 | 0.053 | 0.112 |
| Target language * T–M consistency | 1.576 | 0.218 | 0.047 |
| Target language * T–M consistency * group | 9.855 | 0.004 | 0.235 |
| **Primed, two streams** | | | |
| Test | $F (1,32)$ | $p$ | Partial $\eta^2$ |
| Target language | 2.703 | 0.110 | 0.078 |
| Target language * group | 76.733 | <0.001 | 0.706 |
| T–M consistency | 36.841 | <0.001 | 0.535 |
| T–M consistency * group | 1.260 | 0.270 | 0.038 |
| Target language * T–M consistency | 1.752 | 0.195 | 0.052 |
| Target language * T–M consistency * group | 0.115 | 0.737 | 0.004 |

Group (Native German listeners, $n = 17$; native Chinese listeners, $n = 17$); ANOVA for repeated measures with the factors Target language (German, Chinese) and target-masker (T–M) consistency (target and masker consistent or inconsistent language; two-stream conditions only).

The comparison of native with foreign speech (i.e., speech in a language that is unknown to the participants), both masked by noise, revealed a trend for higher neural activity and better reconstruction of MEG signals for the foreign speech[34]. This rather unexpected result was explained with higher neural variability for the native language, because of deeper processing. When the intelligibility of speech is acoustically degraded by noise vocoding, leaving the envelope intact, a reduction of the neural response locked to the amplitude modulation was observed[35,36]. While this effect was initially suggested to be related to semantic and syntactic contexts[35], it was later argued that at least part of the effect can be explained based on the change of the speech's acoustic fine structure[36]. Whether an additional, language-specific speech processing effect is present on top[37,38] remains disputed[39]. In other contexts, native vowels evoked stronger mismatch negativity (MMN) in the auditory cortex than non-native vowels[40] and enhanced sustained fields have been observed for native vowels and meaningful pitch glides[41]. While the latter studies indicated that language-specific representations may

enhance activity in the auditory cortex, studies using natural speech so far do not confirm this finding[34,42,43].

Here, we explored the role of native language features for the segregation of two speech streams in a doubled-speaker setting: to reduce the influence of basic low-level acoustic cues, we employed a two-speech-stream design with recordings of a single bilingual speaker with native-like competence in both languages. This setup was expected to force listeners to strongly rely on language-specific top-down cues for the segregation task, such as semantic and language-specific phonemic, syllabic, and intonational characteristics, instead of solely attending to speaker-specific acoustic cues, such as pitch and timbre.

Our hypothesis was, first, that the segregation of two speech streams of different languages was easier than the segregation of two streams of the same language produced by an identical speaker. Second, we expected a perceptual advantage for the listeners' native language. Third, we expected that these two effects would be reflected in higher amplitudes of the Nd wave in the auditory cortex[30]. As a previous study with a same-speaker setup had found an attention effect only when the target stream was primed with the isolated target stream[33], we added a comparable condition to our setup.

## Results
### Behavioral results
During the experiment, participants heard two speech streams that were presented in parallel for 6 s. The target-masker-stream combinations were either consistent, i.e., both Chinese or both German or inconsistent, i.e., one stream in Chinese and one stream in German. The stimulus materials in both languages were grammatically meaningful sentence segments taken from five biographies. The target stream was started 1 s prior to the masker stream, and listeners were instructed to focus on this cued stream. They had the task to press the response button each time they detected the target word "Jack" within the target stream. In the second half of the experiment, the primed condition was presented, where the complete target stream was presented alone directly before the presentation of the mixture.

Mean hit and false alarm rates are shown together with the mean sensitivity index ($d'$) across all conditions and separately for both listener groups in Fig. 1; only false alarms in response to the target word in masker streams are reported, which accounted for the majority of false alarms. As an indicator for how well listeners could follow the target stream, $d'$ was generally higher for inconsistent compared to consistent target-masker combinations (unprimed: $F_{32} = 250.377$, $p < 0.001$; primed: $F_{32} = 36.841$, $p < 0.001$; cf. Table 1). There was also an advantage for target streams in the native language, as reflected by an interaction of the target language and listener group (unprimed: $F_{32} = 64.729$, $p < 0.001$; primed: $F_{32} = 76.733$, $p < 0.001$). Finally, $d'$ was significantly higher for primed trials ($F_{32} = 172.327$, $p < 0.001$). The higher $d'$ for primed trials was caused by higher hit rates as well as lower false alarm rates. For consistent-language

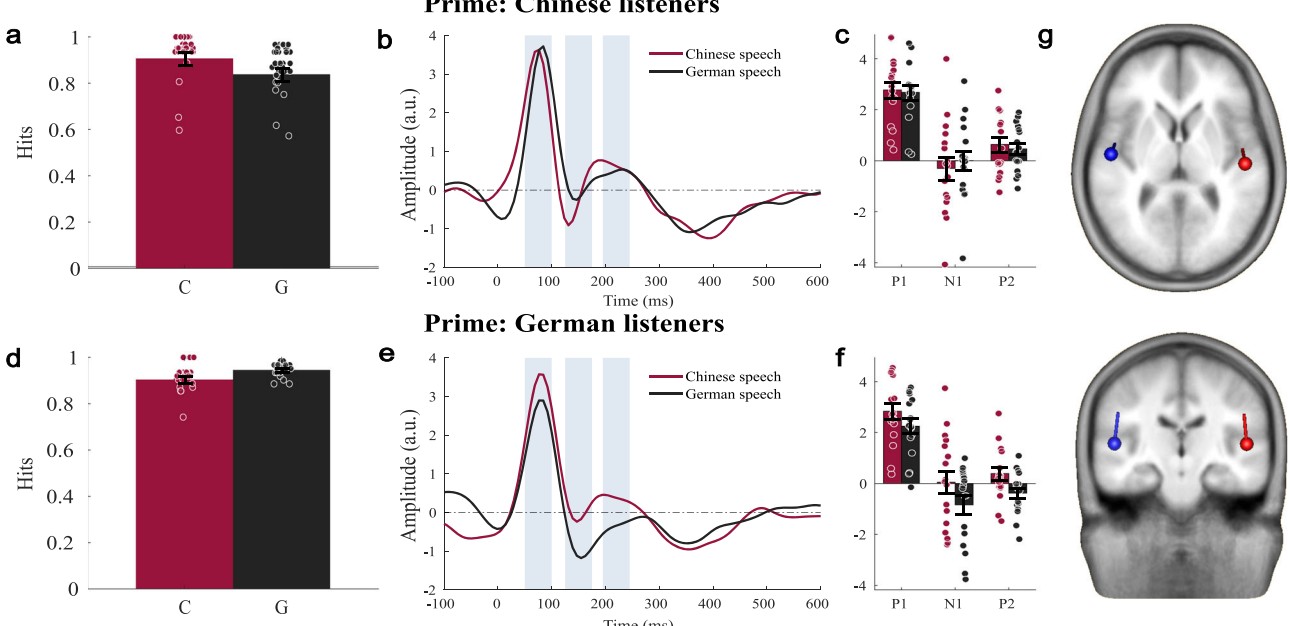

**Fig. 2 | Results for the unmasked, single speech streams used for priming. a, d** Hit rate of Chinese ($n = 17$) and German ($n = 17$) native listeners for the target word in the prime (mean ± standard error; circles indicate the single-subject data). **b** TRF source waveforms for Chinese listeners and **c** the corresponding amplitude of P1, N1, and P2 (mean ± standard error; circles indicate single-subject data). **e** TRF source waveforms for German listeners and **f** the corresponding amplitude of P1, N1, and P2 (mean ± standard error; circles indicate single-subject data). Data related to Chinese speech stimuli are plotted in red, and those related to German speech stimuli are plotted in black. The time windows used to evaluate the P1, N1, and P2 are shaded in gray in the source waveforms. **g** Average dipole positions across all listeners ($n = 34$) fitted to the P1 projected on a standard brain. The individual dipoles were used to generate the source waveforms shown in panels (**b**) and (**e**), as well as in Figs. 3 and 4. The source waveforms are based on dipoles fitted to the P1, left and right hemispheres combined.

streams without priming, $d'$ was never significantly larger than zero, suggesting that listeners were not able to segregate these streams. Note, however, that the low and sometimes negative d' was caused mainly by the high false-alarm rates for consistent-language combinations in the unprimed condition (Fig. 1). Results from the analysis of response times align with results from the analysis of d' (Supplementary Figs. 1 and 2; Supplementary Table 1), with the only difference that no significant effect of target-masker consistency emerged. Finally, a native-language effect was found for the hit rate in isolated, single-stream primes (stimulus language * listener group, $F_{1,32} = 11.608$, $p = 0.002$; Fig. 2a, d).

**Temporal response functions for single speech streams**
The MEG data were evaluated based on the reconstructed temporal response functions (TRFs) at the source level; TRF estimation was based on the first derivative of the envelope of each speech stream, which models the onset response with a focus on the syllable level[44,45]. Target events and button presses were modeled together with the envelope, to reduce their influence on the target-stream response[46].

First, we evaluated the unmasked speech streams, which were used to prime the masked target streams in the second half of the experiment. The source locations were individually fitted to the P1 evoked by the unmasked stimuli, and the same, individual sources were used across all conditions (Fig. 2). Dipoles were generally located in the auditory cortex (Fig. 2g; Supplementary Fig. 3) with similar locations for both groups (Supplementary Table 2).

The source waveforms of these unmasked stimuli showed a positive-going peak with an average latency of 90 ms (±3 ms bootstrap-based standard deviation) and a smaller, negative-going peak at 150 ms (±4 ms bootstrap-based standard deviation), which we refer to as P1 and N1. A second positive-going response, referred to as P2, was present in three out of four grand-average waveforms with an average peak latency of 219 ms (±12 ms bootstrap-based standard deviation). Thereafter, the waveform was negative going in a range of about 300–450 ms. Since no hemisphere main

effects or interactions with hemisphere were observed, the average of left- and right-hemisphere activity is reported below.

The statistical analysis for speech in quiet was based on 50-ms-long, non-overlapping time windows around the peaks as defined in the grand average (P1 50–100 ms, N1 125–175 ms, and P2 195–245 ms, Supplementary Fig. 4). First, we compared whether the responses evoked by German and Chinese speech stimuli differed in one of the time intervals. The comparison revealed a larger P1 ($F_{1,32} = 13.630$, $p < 0.001$) and P2 ($F_{1,32} = 14.61$, $p < 0.001$) for Chinese compared to German speech stimuli in both groups. No significant difference for stimulus language was observed in the N1 time windows ($F_{1,32} = 0.652$, $p = 0.425$).

To evaluate possible reasons for this language effect, we re-analyzed the characteristics of the speech stimuli. Results showed that the number of peaks in the first derivative of the envelope, was somewhat higher for German compared to Chinese in our stimuli. There were 9.5% more envelope peaks in German compared to Chinese within stimulus sequences of the same duration (11,169 onsets in German and 10,199 onsets in Chinese), resulting in an estimated envelope-peak rate of 7.9/s in German and 6.6/s in Chinese. The syllable rate amounted to 5.4/s in German and 4.2/s in Chinese, confirming that the envelope peak rate and syllable rate are proportional to each other. Conversely, the mean amplitude across all envelope peaks was 1.1% higher in Chinese compared to German. This and other physical differences might have resulted in small loudness differences between German and Chinese stimuli despite identical average stimulus power. Loudness was therefore compared using a computational model[47], which suggested that German stimuli (39.2 sones) were about 1.85 sones louder than Chinese stimuli (37.3 sones). This loudness difference is small, and its direction cannot explain the higher P1 and P2 amplitudes for Chinese compared to German stimuli.

Next, we explored if the TRF depended on the native language of the participants. Indeed, a significant stimulus-language * listener-group interaction was observed in the N1 and P2 intervals (P1: $F_{1,32} = 0.390$, $p = 0.537$; N1: $F_{1,32} = 19.356$, $p < 0.001$; P2: $F_{1,32} = 4.655$, $p = 0.039$). The

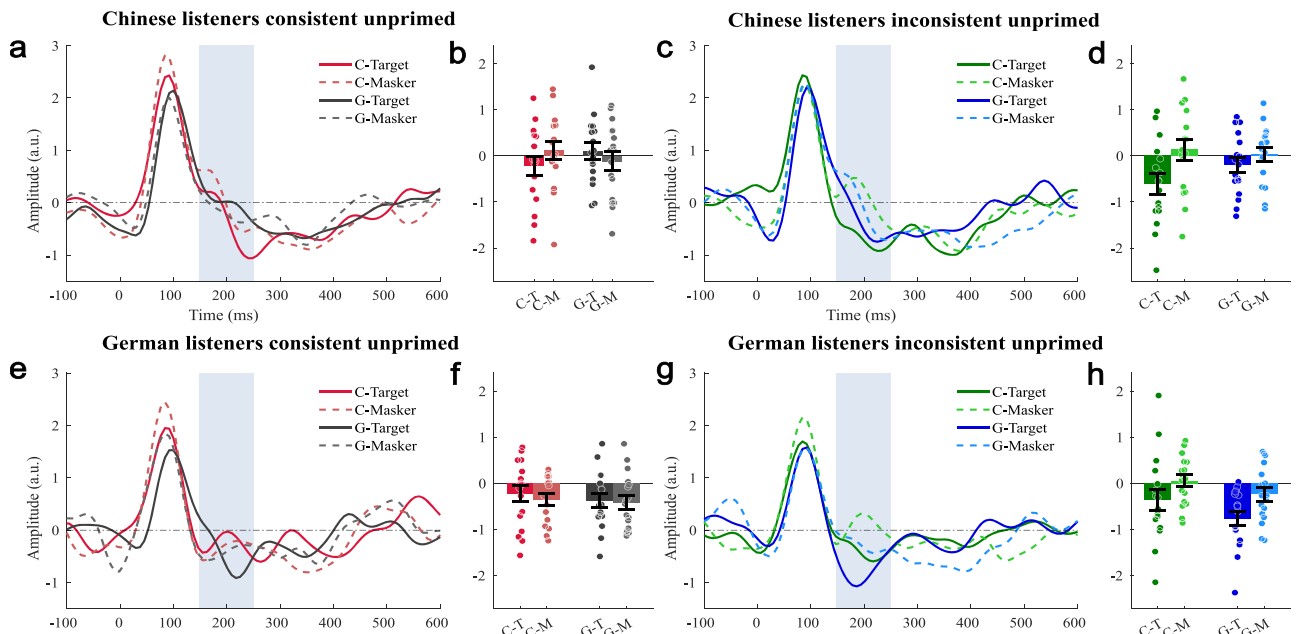

**Fig. 3 | Temporal response function (TRF) in the auditory cortex for masked speech streams in the unprimed condition.** The source waveforms are based on dipoles fitted to the P1 of the single-stream prime, left and right hemispheres combined. Activity evoked by target speech streams is plotted with solid lines, and activity evoked by masker speech streams is plotted with dashed lines. The amplitude for the target is plotted to the left in a darker color and the amplitude for the masker is plotted to the right in a brighter color. The difference between the two corresponds to the negative difference wave (Nd). Note that the control condition is in the same language and was presented in the same stimulus in the consistent stimuli, and in a different condition for the inconsistent stimuli. **a** TRF source waveform for Chinese listeners for unprimed, consistent conditions. **b** Mean amplitude (±standard error; $n = 17$) in the time interval 150–250 ms (gray shading) of the TRF source waveforms shown in panel **a**. **c** TRF source waveforms for Chinese listeners for unprimed, inconsistent conditions. **d** Mean amplitude (± standard error; n = 17) in the time interval 150–250 ms (gray shading) of the TRF source waveforms shown in panel (**c**). **e** TRF source waveforms for German listeners for unprimed, consistent conditions. **f** Mean amplitude (±standard error; $n = 17$) in the time interval 150–250 ms (gray shading) of the TRF source waveforms shown in panel (**e**). **g** TRF source waveforms for German listeners for unprimed, inconsistent conditions. **h** Mean amplitude (±standard error; $n = 17$) in the time interval 150–250 ms (gray shading) of the TRF source waveforms shown in panel (**g**). Circles in panels **b**, **d**, **f**, and **h** indicate single-participant data.

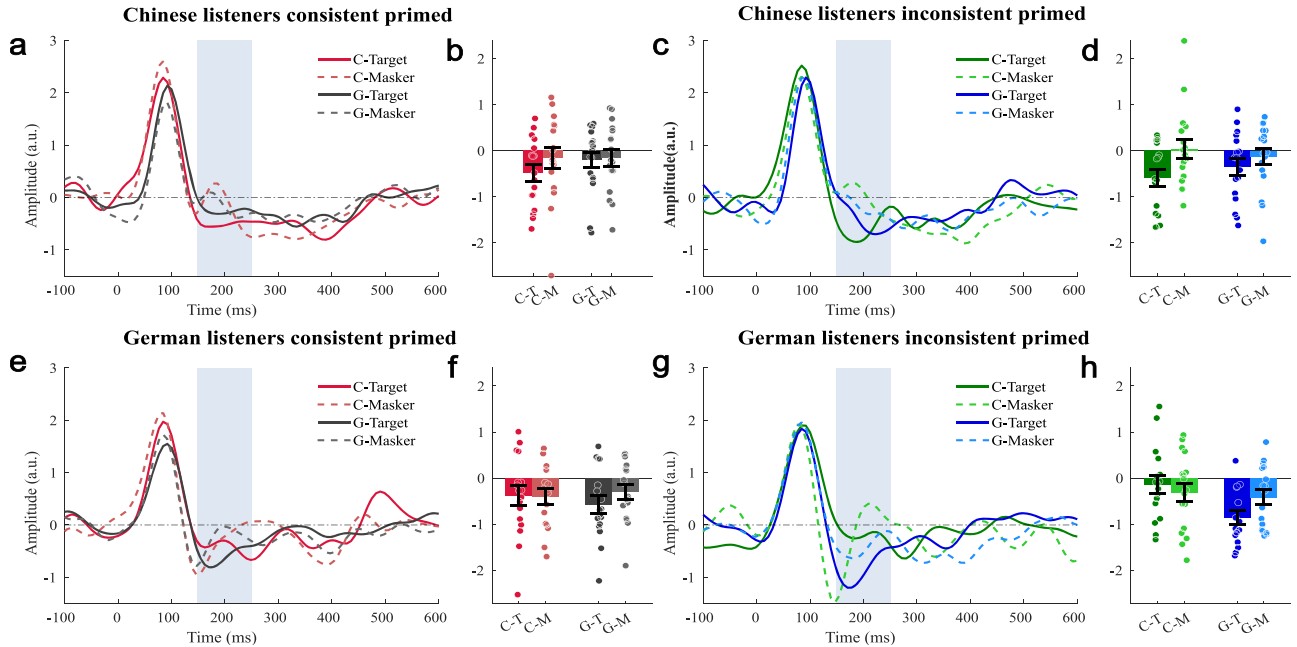

**Fig. 4 | Temporal response function (TRF) in the auditory cortex for masked speech streams in the primed condition.** The figure is similarly arranged as Fig. 3; see there for details. **a** TRF source waveform for Chinese listeners for unprimed, consistent conditions. **b** Mean amplitude (±standard error; $n = 17$) in the time interval 150–250 ms (gray shading) of the TRF source waveforms shown in panel (**a**). **c** TRF source waveforms for Chinese listeners for unprimed, inconsistent conditions. **d** Mean amplitude (±standard error; $n = 17$) in the time interval 150–250 ms (gray shading) of the TRF source waveforms shown in panel (**c**). **e** TRF source waveforms for German listeners for unprimed, consistent conditions. **f** Mean amplitude (±standard error; $n = 17$) in the time interval 150–250 ms (gray shading) of the TRF source waveforms shown in panel (**e**). **g** TRF source waveforms for German listeners for unprimed, inconsistent conditions. **h** Mean amplitude (±standard error; $n = 17$) in the time interval 150–250 ms (gray shading) of the TRF source waveforms shown in panel (**g**). Circles in panels **b**, **d**, **f**, and **h** indicate single-participant data.

direction of the difference was such that the response was more negative-going for native-language stimuli, i.e., the response was stronger for native stimuli in the N1 time window, and weaker in the P2 time windows. This dependence of the N1 on native language was not initially expected for single, unmasked speech streams[34], as our hypothesis was that a difference would emerge under competition when selective attention was deployed successfully to a target speech stream.

## Temporal response functions for competing speech streams

The TRFs of competing speech streams were evaluated in source waveforms based on the P1 sources fitted to the unmasked speech streams (Fig. 2c). In the TRF source waveforms, the P1 was again the most prominent peak (Figs. 3 and 4), but with a somewhat longer response latency of 100 ms (±4 ms bootstrap-based standard deviation) in the across-condition average. The subsequent negative-going response was more variable and, on average, sustained up to about 400 ms. For the analysis of attention effects, the Nd time window was defined based on the grand average across all target stream conditions (Supplementary Fig. 4). The time interval was chosen after the P1 and covering the first peak of the negative-going response in the time range 150–250 ms. An analysis of data reliability in the Nd time interval, based on Cronbach's alpha calculated across participants and all 16 conditions[48], revealed satisfactory values of 0.953 for Chinese and 0.918 for German listeners. For the attention effect in the Nd time interval, no significant main effects of or first-order interactions with hemisphere were observed, such that the statistics below do not report any hemisphere effects for clarity.

## Attention–language interactions for speech mixtures

For the statistical analysis of the attention effect reflected by the Nd, target- and masker-stream-evoked TRFs were compared between stimuli of the same language (Chinese or German) and target-masker consistency (identical language or different language). The analysis of the unprimed conditions (Fig. 3; Table 2) showed a significant main effect of attention across sub-conditions and listener groups ($F_{1,32} = 6.441$, $p = 0.016$). The expected stronger attention effect for inconsistent compared to consistent target-masker combinations is reflected by a significant attention * target-masker consistency interaction ($F_{1,32} = 27.197$, $p < 0.001$); there was also a main effect of consistency ($F_{1,32} = 4.423$, $p = 0.043$). The main hypothesis of stronger attention effects (Nd) for the native language is indicated by a significant attention * stimulus-language * listener-group effect ($F_{1,32} = 5.918$, $p = 0.021$). There was also a significant stimulus-language * listener-group interaction ($F_{1,32} = 4.423$, $p = 0.043$), indicating stronger negative-going responses for native compared to non-native stimuli.

## Effect of priming

The effect of priming was evaluated by comparing the primed with the unprimed data sets in one ANOVA by including priming as an additional factor (Supplementary Table 3). This analysis showed a main effect of priming, caused by overall somewhat more negative-going activity in the 150–250-ms, Nd time window ($F_{1,32} = 9.434$, $p = 0.004$) for the stimuli in primed conditions. The primed and unprimed stimuli were presented block-wise, with the unprimed block presented second. Thus, an additional effect of practice in addition to priming cannot be excluded.

Based on a previous study[33], an effect of priming on attention was expected, showing a priming * attention interaction, which, however, was not observed for the Nd ($F_{1,32} = 0.030$, $p = 0.864$). Because there was a significant priming * attention * target-masker-consistency interaction ($F_{1,32} = 6.074$, $p = 0.019$), we also tested the data from consistent and inconsistent conditions separately, but did neither find a significant priming * attention interaction for consistent-language ($F_{1,32} = 3.031$, $p = 0.091$) nor for inconsistent-language mixtures ($F_{1,32} = 2.444$, $p = 0.128$). No further interaction with priming was observed.

The attention * stimulus-language * listener-group interaction ($F_{1,32} = 11.127$, $p = 0.002$) and the stimulus-language * listener-group

**Table 2 | Statistical analysis of two-stream TRF source waveforms in the Nd time window**

| Unprimed | | | |
|---|---|---|---|
| Test | $F$ (1,32) | $p$ | Partial $\eta^2$ |
| attention | 6.441 | 0.016 | 0.168 |
| attention * group | 0.194 | 0.663 | 0.006 |
| stimulus language | 0.665 | 0.421 | 0.020 |
| stimulus language * group | 4.423 | 0.043 | 0.121 |
| T–M consistency | 1.054 | 0.312 | 0.032 |
| T–M consistency*group | 1.703 | 0.201 | 0.051 |
| attention * stimulus language | 2.914 | 0.097 | 0.083 |
| attention * stimulus language * group | 5.918 | 0.021 | 0.156 |
| attention * T–M consistency | 27.197 | <.001 | 0.459 |
| attention * T–M consistency*group | 0.381 | 0.542 | 0.012 |
| stimulus language * T–M consistency | 0.443 | 0.511 | 0.014 |
| stimulus language * T–M consistency*group | 3.269 | 0.08 | 0.093 |
| attention * stimulus language * T–M consistency | 0.051 | 0.823 | 0.002 |
| attention * stimulus language * T–M consistency * group | 0.002 | 0.969 | 0.000 |
| **Primed** | | | |
| Test | $F$ (1,32) | $p$ | Partial $\eta^2$ |
| attention | 5.260 | 0.029 | 0.141 |
| attention * group | 0.886 | 0.354 | 0.027 |
| stimulus language | 1.175 | 0.286 | 0.035 |
| stimulus language * group | 6.500 | 0.016 | 0.169 |
| T–M consistency | 0.037 | 0.848 | 0.001 |
| T–M consistency * group | 0.000 | 0.987 | 0.000 |
| attention * stimulus language | 0.213 | 0.648 | 0.007 |
| attention * stimulus language*group | 11.127 | 0.002 | 0.258 |
| attention * T–M consistency | 0.948 | 0.338 | 0.029 |
| attention * T–M consistency * group | 0.938 | 0.34 | 0.028 |
| stimulus language * T–M consistency | 3.028 | 0.091 | 0.086 |
| stimulus language * T–M consistency * group | 0.952 | 0.337 | 0.029 |
| attention * stimulus language * T–M consistency | 0.174 | 0.68 | 0.005 |
| attention * stimulus language * T–M consistency * group | 0.952 | 0.337 | 0.029 |

Group (Native German listeners, $n = 17$; native Chinese listeners, $n = 17$); ANOVA for repeated measures with the factors attention (target versus masker stream of the same language), stimulus language (German, Chinese), and target-masker (T–M) consistency (target and masker consistent or inconsistent language; two-stream conditions only).

interaction ($F_{1,32} = 6.500$, $p = 0.016$) were independently reproduced in the primed data set (Table 2). In contrast, the attention * target-masker-consistency interaction found significant for unprimed trials, was not significant for primed trials ($F_{1,32} = 0.948$, $p = 0.338$). This difference between the primed and unprimed data sets is also reflected in the priming * attention * consistency interaction (Supplementary Table 3). Even though the direct comparison showed only the non-significant trend, it appears that

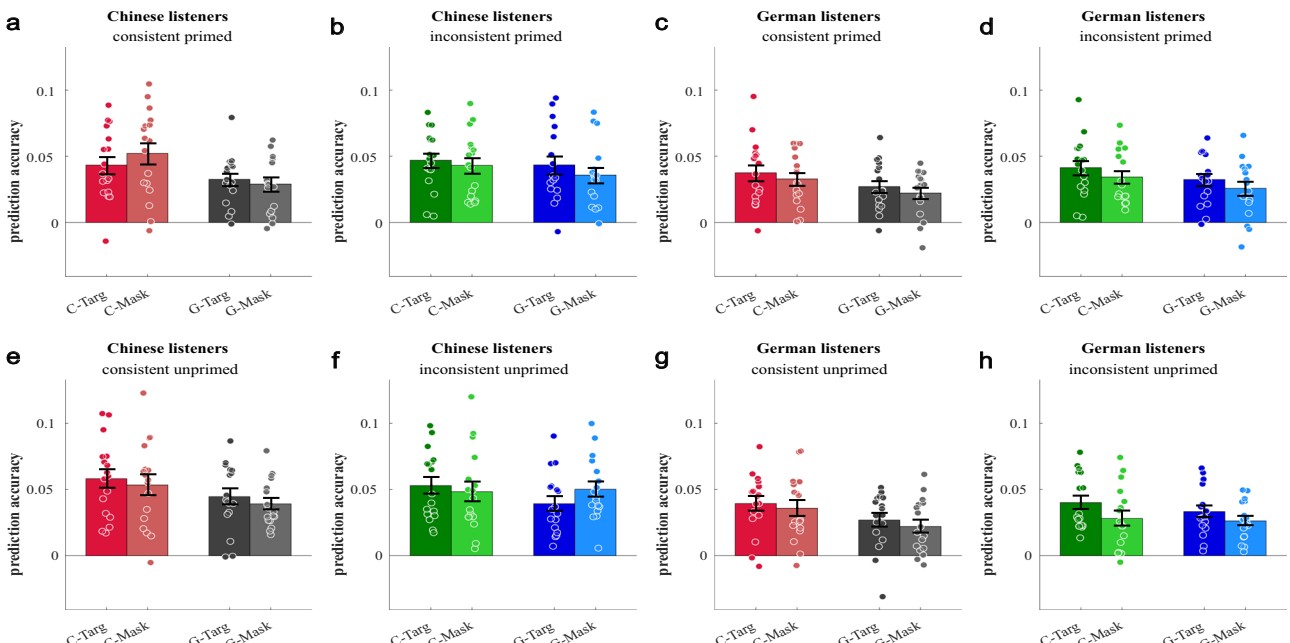

**Fig. 5 | Neural encoding of MEG source activity in the auditory cortex based on the full TRF.** The measure indicates how well the continuous MEG activity can be estimated based on the envelope when convoluted with the TRF estimated in another data set. Each single-participant data (small circles) is based on an average of five data segments evaluated in a leave-one-out cross-validation (bars indicate mean ± standard error across participants; $n = 17$ per group). In each pair of bars, the left (darker color) represents the target, and the right (brighter color) the masker of the same language and subcondition. **a** Encoding accuracy for Chinese listeners for primed, consistent conditions. **b** Encoding accuracy for Chinese listeners for primed, inconsistent conditions. **c** Encoding accuracy for German listeners for primed, consistent conditions. **d** Encoding accuracy for German listeners for primed, inconsistent conditions. **e** The encoding accuracy for Chinese listeners for unprimed, consistent conditions. **f** Encoding accuracy for Chinese listeners for unprimed, inconsistent conditions. **g** Encoding accuracy for German listeners for unprimed, consistent conditions. **h** Encoding accuracy for German listeners for unprimed, inconsistent conditions.

this interaction is primarily driven by somewhat stronger attentional enhancement for consistent, primed trials.

### Neural encoding of individual speech streams

Next, we tested if the TRF results could be confirmed by an encoding analysis, and if an attention effect of priming could possibly be detected. The neural encoding analysis in the present case evaluates how well the ongoing MEG response can be linearly predicted based on the envelope and TRF of a given condition, here for the target and masker streams, respectively (Fig. 5). Stronger TRF responses typically result in higher prediction accuracy, also referred to as enhanced "neural tracking" of the stimulus.

Consistent with the TRF results, the encoding analysis (Fig. 5; Supplementary Table 4) confirmed a main effect of attention across unprimed and primed conditions ($F_{1,32} = 8.564$, $p = 0.006$). However, the data did not confirm the attention * stimulus-language * group interaction ($F_{1,32} = 0.073$, $p = 0.789$), the attention * priming interaction ($F_{1,32} = 0.015$, $p = 0.903$), or the attention * stimulus-consistency interaction ($F_{1,32} = 0.650$, $p = 0.426$). Conversely, the encoding analysis showed a significant effect of the stimulus language ($F_{1,32} = 32.785$, $p < 0.001$; stimulus-language * consistency: $F_{1,32} = 5.686$, $p = 0.023$), caused by stronger encoding for Chinese compared to German speech stimuli.

The encoding analysis reported above was based on the complete TRF waveform. To understand the source of the discrepancy between the TRF and encoding analyses, we performed the analysis separately for the P1 and Nd time windows (Supplementary Figs. 5 and 6): This analysis revealed that the total TRF analysis (Fig. 5) is dominated by activity in the P1 time window (Supplementary Fig. 5). While in the Nd time window, stronger tracking for German target streams was observed in the group average of native German speakers, stronger tracking for Chinese masker streams was observed in the group of native Chinese speakers. When the latter condition was sorted into two subgroups (Supplementary Fig. 7), one with stronger target and the other with stronger masker tracking, the TRFs revealed that the group with

stronger masker tracking showed a strong P2 for masker streams, which was weaker for target streams, explaining the results of the encoding analysis. On the other hand, the group with stronger target encoding showed a predominantly negative-going wave in the Nd time window, which was stronger for target stimuli. For both subgroups, however, the TRF was consistently more negative-going for target compared to masker streams in the Nd time interval. This subgroup analysis thus demonstrates that the P2 and Nd components interact non-linearly in the Nd time window and that Nd amplitude, therefore, cannot be linearly translated into enhanced prediction accuracy in the encoding analysis. As such, the discrepant results found in the encoding analysis do not compromise the results obtained in the TRF analysis.

### Discussion

These results confirm the hypothesis that focusing on a speech stream is associated with stronger attention-related activity in the auditory cortex when the target speech is in the native language than in a foreign language. Second, stronger attentional enhancement was observed when speech streams were made up of a native and a foreign language. Enhanced negative-going activity for speech in the native language was not only observed under competition between two speech streams, as originally expected but already when a single speech stream was presented in quiet.

The finding of higher TRF amplitudes of the P1 and P2 for Chinese compared to German speech is most likely related to the lower envelope-peak rates of our Chinese stimuli. The envelope-peak rate is loosely related to the syllable rate, and a previous report found higher amplitude responses with lower syllable rate[49]. It seems unlikely that the difference in envelope-peak rate in our stimuli is accidental. A recent meta-analysis confirms that syllable frequencies in German are higher than in Chinese during reading aloud[50]. While it is possible that our speaker was more proficient in German compared to Chinese, the syllable frequencies measured for our stimuli are in the range reported previously for both languages[50]. An additional factor

for higher peak rates could be German-specific consonant sequences[51], which are absent in Chinese[52], and which could produce additional envelope peaks within syllables. The alternative explanation that a higher perceived loudness of the Chinese stimuli contributes to the effect, which may be due to language-specific differences, was not supported by a computational loudness model.

In the present data, the main effect of language and supposedly envelope-peak rate was only observed for the positive-going components. Because the Nd (i.e., attention) analysis was based on a contrast within the same language, it should not be influenced by the language-related amplitude difference. If the repetition rate also had a significant influence on the Nd, the analysis should have produced an attention * stimulus-language effect independent of the listener group, which was not observed in the TRF analysis.

Even though the task was identical across conditions, task difficulty strongly increased when a masker was present, and more so when the masker was in the same language as the target. The difficulty of segregating target and masker streams is reflected in low hit rates and high false alarm rates. The d′ for unprimed conditions was only positive and significantly different from zero when the masker-target combination was inconsistent, whereas segregation of target words in target and masker streams was not possible when both streams were in the same language, i.e., in the consistent combination. Note that the possibility of confusing the two streams was also highest for unprimed consistent streams. However, dissociating the effects of confusion and segregation is difficult in the consistent stream condition. Regarding our main hypothesis it is important that there was an advantage for the detection of the target word within streams of the listeners' native language[18]. Priming enhanced d′ across conditions by increasing hit rates and decreasing false alarm rates. Thus, it is possible that the segregation of speech streams was indeed supported by priming. Note that the target words occurred in the same position in unmasked (prime) and masked streams. Theoretically, enhanced performance could result from detecting and remembering the approximate temporal occurrence of targets during the priming phase and then selecting valid targets during the two-stream presentation based on this memory. The detection of a target word was used in the present experiment to increase reliance on the use of all available linguistic information. A previous study with consistent speech stimuli used silent pauses as targets, which could only be presented in the two-stream presentation phase and not in a priming phase[33]. This procedure resulted in a similar enhancement of performance, supporting the interpretation of the current results to indicate at least some reliance on segregation mechanisms rather than just reliance on prior target information.

The stronger Nd response observed here for native-language stimuli is likely related to a more precise deployment of attentional resources toward the target stream. A high level of experience with the linguistic features of the language of the speech stream allows for a better prediction of subsequent speech sound elements, which will be required for attentional deployment in the absence of strong low-level segregation cues such as differences in spectral envelope or space. A similar mechanism had previously been suggested based on the comparison of natural and noise-vocoded speech[35]. However, subsequent studies suggested that this comparison may reflect the difference in the stimulus fine structure rather than genuine speech perception[36,39]. A number of further studies explored if the processing of noise-vocoded speech in the auditory cortex is enhanced when listeners receive the content beforehand (priming), but have so far revealed inconsistent results[53]. Thus, previous studies could not clarify whether language-specific speech cues can be used to enhance (attentional) processing in the auditory cortex.

The present study differs from previous studies in two critical respects. First, the task was the segregation of two speech streams, not the perception of a single, distorted stream. Second, we compared two groups of listeners with different native languages, making different predictions for native and non-native speech stimuli. We argue that the present TRF data support the role of language-specific cues for attentional enhancement in the auditory cortex.

Ongoing speech from one speaker is typically grouped into an auditory stream[4], with similar qualities as objects in vision[54]. By combining two acoustical speech streams from a single speaker, stream segregation based

on non-linguistic cues is almost impossible, such that top-down attention plays a major role in selecting the information of interest. Basic linguistic cues may provide some cues for streaming[55] in inconsistent speech mixtures, but hardly in consistent speech mixtures in the present setup. Priming may then provide information on how to deploy attention selectively to the mixture, and this might even be possible when the two speech streams cannot be perceptually segregated in their continuity if certain transients are selectively enhanced by attention[56].

In the setup used in this study, priming did not significantly increase the attentional enhancement of the target over masker streams, which had been observed in a previous study[33], but we found a trend for such an enhancement for consistent speech stimuli. Thus, our findings do not contradict the findings from the previous study. The difference might be due to the fact that the previous study only evaluated a consistent-stream combination and thus had higher power to detect an effect. However, the present data additionally showed a general response enhancement of negative-going activity, i.e., for target and masker streams alike. While such an effect would not be predicted by selective attention, it may still be relevant for segregation. Studies that investigated stream segregation based on tone frequency report an enhancement of both streams caused by lesser mutual suppression in the auditory cortex[10,12]. One might speculate that the negative-going response enhancement observed for priming could be related to a top-down segregation of the two streams that could support the task even without stronger selective enhancement of the target stream[54], although this effect has rather been observed for the P1[57,58]. Relative enhancement of distractor streams has been observed previously for cued over uncued listening to multi-speaker scenes[31] or when changing the target during stimulus presentation[44]. It is thus unclear, if the enhancement of both streams is caused by a lack of selectivity, because of insufficient stream segregation, or could be a sign of successful top-down segregation. A potential approach to decide this would be the comparison based on the performance in individual trials, but the present data do not provide enough trials per condition to evaluate such a sub-stratification. Finally, since unprimed blocks always preceded primed blocks in the current study, a practice effect can also not be ruled out completely.

The encoding analysis was dominated by the P1, which can explain why encoding was stronger for Chinese compared to German speech stimuli (cf. discussion of language differences above). However, we did not reproduce the native-language advantage observed for the Nd in the encoding analysis. The discrepancy between the TRF and the encoding analyses (in the Nd time window) most likely lies in the interaction of P2 and Nd, such that stronger encoding accuracy can either be observed as a more pronounced Nd for target trials, or stronger P2 for masker trials, with considerable inter-subject variability. Critically, stronger Nd for target streams was also observed for participants who showed strong P2 coupled with stronger neural tracking for masker stimuli. The negative encoding analysis, therefore, does not contradict the TRF-based results discussed above. Most studies on selective attention for both tone[59,60] and speech streams[22,31,58,61] report enhancement of negative-going activity in the auditory cortex. Note, however, that other EEG studies have also reported enhanced P2 for attended speech streams[46,62] and higher consistency between encoding and TRF results. It remains unclear, which details in the stimuli or analysis strategy cause these differences in P2 effects between studies.

Most previous studies investigated a smaller number of different conditions and accordingly had longer speech intervals to estimate TRFs than the 3 min per condition available in the present study. A recent analysis of the influence of data quantity on TRF estimates[63] demonstrated high consistency between TRFs obtained in a range from 2 to 41 min. However, in an encoding analysis, prediction accuracy was not different from zero for a few subjects unless the data were longer than 15 min. For the present data, we obtained good reliability in the Nd time window based on Cronbach's alpha, and the main TRF results were reproduced in the second, primed stimulus set. We, therefore, think that data reliability is sufficient for the claims made regarding native-language and target-masker consistency

based on the TRF analysis. Possibly, the encoding analysis would have had more power to investigate the effects of native language if longer speech intervals had been available per condition, but it is not to be expected that this would have changed the non-linear interaction between Nd and P2 reported above.

Our hypothesis for enhanced Nd in native language trials was based on the assumption that prediction of the speech stream allows for more efficient deployment of attention. An alternative explanation could be that language-specific processing is already present at the level of the auditory cortex and could be the source of the enhanced negative-going activity for native-speech stimuli. Such an interpretation might be supported by the higher negative-going response observed in the processing of the single-stream priming stimuli in the present study.

A number of studies have suggested that phoneme encoding is implemented early in the auditory cortex[64–66] (but see Daube et al.[44]) and that the negative-going response enhancement observed in two-speaker scenes in fact represents a selective activation of phoneme encoding[67]. Moreover, two recent studies demonstrated that phoneme encoding in a second language (L2) changes with language proficiency[68,69]. It is therefore conceivable that phonemic differences between German and Chinese[41] are differently represented depending on native language and that this difference is a source of the native-language effect observed in the present study. However, when we tried to test this hypothesis with phoneme modeling, we found that the percentage of non-overlap phonemes is small, reducing the signal-to-noise ratio to an extent that further analyses were not feasible. Moreover, previous data indicate lower phoneme-TRF amplitudes for more proficient L2 speakers[68], making it less likely that larger amplitudes for native peak onsets are strongly influenced by phonemic processing. The large phonemic overlap between the languages also makes it unlikely that the difference observed is related to phonemic processing alone rather than to differences in attention. A direct comparison between native and non-native speech in previous studies produced mixed results[70]. One study did not find higher amplitudes for native speech[34]. This could be related to the higher similarity between the languages used in the latter case, or to differences in participants' tasks: In the study that found stronger amplitudes for foreign speech, the task was to detect the repetition of a single syllable, while the task in the present study was the detection of a word (the name Jack). While syllable detection may only, to some extent, benefit from linguistic context, the detection of a name can be supported by the syntactic and semantic environment to a much greater extent, given sufficient linguistic experience, which, in our case, only native speakers possess. Another study that investigated text comprehension found stronger neural-encoding accuracy for native (L1) Chinese compared to L2 English language stimuli. This effect might be related to speech proficiency, although the listeners' considerable proficiency in English makes a direct comparison to our study difficult. Moreover, it is possible that a similar language-dependent difference between Chinese and English explains these results, as found here for Chinese and German independent of native language. While semantic context can influence early activity in the auditory cortex[71], it appears that this is not reflected at the level of the envelope following response[72]. Whether such semantic cues could be relevant for the stronger negative-going response in native speakers in the present study requires further evaluation.

## Materials and methods
### Participants
Seventeen Chinese native speakers between the ages of 24 and 34 (10 female) who had little knowledge of and little exposure to German and 17 native speakers of German between the ages of 19 and 33 (10 female) with no prior knowledge of Chinese were recruited. 13 of the Chinese participants had participated in a one-month German course before moving to Germany, but none of them reported to understand German when they participated in the experiment. The average time that Chinese participants had spent in Germany was 8.5 months (3–15 months). All participants reported normal

hearing. They were paid for their participation and provided written informed consent. The procedure was approved by the Ethics Committee of Heidelberg University Hospital. All ethical regulations relevant to human research participants were followed.

### Stimuli
Chinese and German text passages, which were chosen from several published biographies, were read by a German–Chinese bilingual speaker at a normal speech rate and recorded in a soundproof, anechoic room. All names of main characters in those biographies were replaced by a target word, which in our case was the name "Jack", being spoken newly each time in the context of the text. This common English name was chosen because it was expected to be known to both Chinese and German listeners alike. Recordings were performed with an RME Fireface sound card via a dynamic AKG microphone with a sampling rate of 44,100 Hz. All pauses in the speech stream longer than 300 ms were shortened to 300 ms. Target and masker streams were first cut into 7-s- and 6-s-long segments, respectively, ramped on and off with 17-ms-long time windows. The intensities of each stream were normalized to equal RMS. Target and masker streams were mixed into one mixture stimulus. In the mixture, the 7-s-long target streams were started 1 s earlier than the 6-s-long non-target streams in order to indicate the targets to the participants. As the target word, the name "Jack" occurred up to two times in each target or non-target stream. Target words in target and masker streams were at least 2 s apart on the timeline. In a subset of the stimuli, the target stream was additionally primed by playing the complete 7-s target stream alone, directly before the mixture.

### Experimental procedures
To test how linguistic competence influenced selective attention, the target speech stream could be either native or non-native speech and was masked by a different, second speech stream, which could also be native or non-native speech. As a result, four conditions categorized by language type of target and masker streams were created. In the first section of the experiment, the stimuli were directly started with the 1-s-long target cue (unprimed). There were 30 trials for each condition (yielding three minutes of speech for the evaluation per condition)[63]. The order in which the four conditions were played was counterbalanced across subjects. In the second section of the experiment, the target streams were presented once as prime before the mixtures. Each mixture was used only once in the whole experiment. Stimuli were DA converted (RME ADI-8 DS), passed through a programmable attenuator (TDT PA5), amplified (TDT HB7), and presented with insert earphones via foam earpieces (ER-3, Etymotics Research, Elk Grove Village, IL, USA. T). Stimuli were presented diotically with an average sound pressure level of 80 dB for the combined speech streams, measured with a Brüel & Kjaer microphone (Ear Simulator Type 4157) in a 2cc coupler.

Participants were instructed to follow the speech stream that started first (the target) and to ignore the masker stream that started second (the masker stream). They were instructed to detect the target word "Jack" in the target streams, but ignore its occurrence in the masker streams. Each stream had 0, 1, or 2 target words with the probability of 10%, 80%, and 10%, respectively. The position of the target word was grammatically correct (it only occurred in positions where names can appear). It was guaranteed that the target did not appear during the first second of the target speech streams. Participants were instructed to press a response button when they detected the target. Only responses in a 2-s time window after target word offsets were registered as hits, and if no button was pressed in the 2-s time window, the targets were considered misses. Button presses in response to the target word presented within the masker stream were considered false alarms. The inter-trial interval was set to 2 s.

### MEG recording and pre-processing
Before the MEG recording, each subject was seated in a chair for digitization and was asked to remain still through the whole procedure. Four head-

position indicator (HPI) coils were attached to the mastoid and temples at each side of the head. Besides those four coil positions, 100 additional positions distributed over the whole head were digitalized with a Polhemus Isotrak system. The position of the HPI coils was then measured at the beginning of the MEG recording. The MEG data were recorded with a Neuromag-122 whole-head system with 1000 Hz sampling rate in direct-coupled mode. For further processing, all channels with excessive noise or severe signal loss were deleted before the analysis. MEG data were then bandpass filtered 1-20 Hz (zero-phase, fourth-order Butterworth filter) and resampled to 128 Hz for further analysis.

## Temporal response functions

The speech stimuli were first bandpass filtered from 200 to 5000 Hz (zero-phase, second-order Butterworth filter). The speech envelope was then extracted by applying a Hilbert transform and subsequent application of a low-pass filter at 10 Hz (zero-phase, second-order Butterworth filter). To focus on onsets within this signal, the first derivative of the speech envelope was calculated, and then half-wave rectified. The latter was used based on the assumption that the auditory cortex response is mostly confined to sound onsets[31,45,49]. For the analysis of the mixtures, the cues—i.e., the first second of the target stimuli—were not included in the analysis.

In addition to the envelope onsets, the onset of the target word ("Jack") in the target and separately in the masker speech streams and the button responses were modeled as three additional regressors to avoid these events were instead explained by the envelope-onset regressors.

The evaluation of speech-envelope-related activity was performed with the mTRF toolbox 2.0[73]. A temporal response function approach was employed to simulate how speech envelope was encoded in the MEG responses assuming that the brain is a linear time-invariant system with stimulus features as input and MEG responses as output.

$$r(n) = \sum T(m)s(n-m) + e(n)$$

Where $r(n)$ was the MEG signal in source space, $T(m)$ the temporal response function (TRF), $s(n)$ the stimulus speech envelope onset, and $e(n)$ the residual error. In order to avoid an ill-posed problem and overfitting, an L2 penalty with the parameter lambda was added and tuned to provide the highest predictive power. The ridge parameter lambda was adjusted using leave-one-out cross-validation by minimizing the prediction error (mean-square error) between the MEG response and the predicted response when single speech streams were played in quiet. The optimal lambda parameter was searched for in the range from $10^{-6}$ to $10^{10}$, yielding values in a range from $10^3$ to $10^6$ ($6 \times 10^3$, $27 \times 10^5$, $1 \times 10^6$). These parameters were then applied to all conditions, to provide comparability of response amplitudes between conditions. The peak latencies of the temporal response functions for speech in quiet were estimated using bootstrapping with 5000 resamples.

## Source localization for TRFs

Source analysis was performed with BESA software (version 6.1), using a spherical head model fitted to the digitized head-surface points. The average of all TRFs for the single speech stream (the prime) was used for dipole fitting. For each subject, two dipoles in bilateral auditory cortex were used as starting solution and then were fitted to the P1 peak. This dipole model was then used as a spatial filter to estimate the source waveforms of the TRFs in auditory cortex from all conditions for each participant. The P1 was chosen because it was the most consistent evoked component across participants and provided the best signal-to-noise ratio.

## Neural encoding analysis

Neural encoding evaluates how well the temporal response function can predict the ongoing MEG based on the derivative of the acoustic envelope after convolution with the single attended and unattended streams. The prediction accuracy was calculated for these two conditions based on the TRF in the full time range from −500 to 1000 ms, by calculating the correlation coefficient between the predicted and the recorded, continuous MEG signal. Additional exploratory analyses were performed for the P1 and Nd time range used for the TRF evaluation (Supplementary Figs. 5 and 6). The analysis was performed in source space, using the two auditory-cortex dipoles fitted to the P1 of the single-stream TRF. For the modeling of the ongoing, raw MEG data, two regional sources in the position of the eyes, and a PCA component modeling slow environmental artifacts (mostly caused by streetcars) were added to the spatial filter. The encoding analysis was performed with a leave-one-out procedure, where the TRF was estimated from 80% of the data and then tested with the remaining 20%. For each data set, this procedure was repeated for five different subsets and the mean encoding accuracy was retained for the group analysis. The encoding analysis was only performed for the two-stream data sets. The lambda values used for the TRF fitting were previously determined with the single-stream data. For estimating the TRFs, the target and masker envelope were modeled together; the additional regressors were not used for this analysis, however, because they mostly modeled activity outside of auditory cortex.

## Statistics

Behavioral results were evaluated using the sensitivity index d', which was derived from hit and false alarm rates. When hit rate equals 1, the form $(n-0.5)/n$ is taken as corrected hit rate. For trials without false alarm, $0.5/n$ is used as false alarm rate for the d' calculation[74]. The single-stream condition was evaluated based on hit rates, because no second stream was available and the false alarm rate was otherwise very low.

The MEG response was measured as average amplitude in the individual source waveforms in time intervals pre-defined based on the grand-average source waveforms across conditions and participants (Supplementary Fig. 3). For unmasked speech stimuli, intervals were chosen for the P1, N1 and P2 in the intervals 50–100 ms, 125–175 ms, and 195–245 ms. These 50-ms long time intervals were chosen to cover the peak and avoid overlap between the three successive peaks. For the masked, speech-mixture conditions, the P1 was measured in the intervals 60–110 ms. The subsequent, negative going response was overall more broad-based than the N1 in unmasked conditions (Figs. 3 and 4). The grand average across all target conditions and groups showed a negative-going response rising subsequent to P1 and sustaining up to 400 ms with superimposed peaks at approximately 210 and 370 ms. A 100-ms-long time interval from 150 to 250 ms was therefore chosen, starting after the P1 and covering the rising part of the negativity including the first superimposed peak in the grand average. The mean amplitudes were evaluated with an ANOVA with the repeated measures attention (2), language (2), target-masker-language consistency of speech-mixtures (2), hemisphere (2), and with the between subject factor native language (2). In this design, target and masker evoked TRFs were compared for the same stimulus language, i.e., within trials for the coherent-language conditions, and between trials for the inconsistent language conditions. The normality of the data for different conditions was tested with the Kolmogrorov–Smirnov test for normality. For component mean amplitudes, all conditions survived the normality test ($p > 0.05$).

## Statistics and reproducibility

The main effects and their interactions were considered significant when the p-value was smaller than 0.05. The effect size was measured with partial Eta squared with the multivariate ANOVAs in JASP[75].

## Reporting summary

Further information on research design is available in the Nature Portfolio Reporting Summary linked to this article.

## Data availability

The behavioral and MEG data are available at heiDATA, the online data repository of Heidelberg University (https://doi.org/10.11588/data/V57GNG)[76].

## Code availability

All code that was customized for the current study and used to create figures is also available at heiDATA (https://doi.org/10.11588/data/V57GNG)[76].

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

## Acknowledgements

This work was supported by the Deutsche Forschungsgemeinschaft grant DFG 593/5-1 (A.G.) and by the China Scholarship Council (M.L.). Publication costs partially covered by Heidelberg University.

## Author contributions

M.L. and A.G. designed the study in consultation with J.G.; M.L. performed the experiments and the data analysis under the supervision of A.G.; J.G. provided critical feedback and helped interpret the analysis; and M.L., A.G., and J.G. drafted the primary paper and edited it.

## Funding

## Competing interests

The authors declare no competing interests.
