## [Transparent Peer Review file · Communications Biology]

A listening advantage for native speech is reflected by attention-related activity in auditory cortex

Corresponding Author: Dr Alexander Gutschalk

Version 0:

Reviewer comments:

Reviewer #1

(Remarks to the Author)
Review

Brief summary of the manuscript:

The author asserts that while the advantage of native speech comprehension is widely acknowledged, the underlying neural mechanisms behind this phenomenon remain elusive. To address this gap, the authors conducted a study employing magnetoencephalography (MEG) to investigate whether attentional enhancement in the auditory cortex is more pronounced for native speech. They utilized bilingual speakers producing Chinese and German speech, creating a simulated cocktail party scenario with both consistent and inconsistent language combinations. Native speakers of Chinese and German participated in a target detection task within the designated speech stream.

The findings revealed that attentional enhancement manifested as increased negative activity in the temporal response function. Notably, this enhancement was more pronounced when the target speech was in the participants' native language compared to a non-native language. Furthermore, it was heightened when presented with inconsistent language combinations compared to consistent ones. The authors suggest that this heightened neural activity for native speech may be attributed to a more accurate top-down prediction mechanism specifically tuned to the native speech stream.

In summary, the study sheds light on the neural underpinnings of the listening advantage for native speech, highlighting the role of attentional mechanisms in facilitating comprehension, particularly in challenging linguistic environments.

Overall impression of the work:

Overall, this manuscript presents a well-executed study investigating the neural basis of the listening advantage for native speech comprehension. The writing is clear and concise, effectively conveying the research aims, methods, and results. The figures provided are visually appealing and aid in understanding the experimental setup and findings.

The study design, incorporating magnetoencephalography (MEG) to examine attentional enhancement in the auditory cortex, is methodologically sound. By utilizing bilingual speakers and creating a realistic cocktail party scenario with both consistent and inconsistent language combinations, the authors were able to simulate real-world listening conditions and capture relevant neural responses.

The results of the study provide valuable insights into the mechanisms underlying native language processing, demonstrating heightened attentional enhancement for native speech compared to non-native speech. Additionally, the finding that attentional enhancement was more pronounced in response to inconsistent language combinations adds nuance to our understanding of language processing in complex environments.

Overall, this manuscript contributes significantly to the literature on language comprehension and attentional mechanisms. The thoroughness of the study design, clarity of presentation, and compelling findings make it a noteworthy addition to the field. While the manuscript is strong, a few additional analyses could further strengthen the work (see comments above).

Specific comments, with recommendations for addressing each comment

Minor comments:

36: Listening to native speech at usual levels...

487: Have their hearing thresholds been measured?

516: Stimuli were presented diotic?

525: Why a 2-s time window? I would suggest a time window that exclude biological implausible early responses for instance (150ms-2000ms, see Bendixen, 2013). A histogram of hits and false alarms could be helpful to set the boundaries of the window.

False alarms were solely related to button presses outside the window of the target word? Maybe it would be worth checking if some of the false alarms are appeared out of nowhere or if some of the false alarms are related to the target word in the masker stream.

97: Was the same target stream presented twice? If so, could be habituation a potential confound?

99-105: Figure 1: would add also the explanation of the wo different bars (primed/unprimed) to figure caption

148: would help to indicate significant windows already in the plot

307: different citation style

381: un-primed unprimed

Major comments:

1. Could the higher TRF amplitudes of the P1 and P2 components also be driven by a difference in perceived loudness between Chinese and German stimuli? Due to its mathematical nature (squaring), the RMS underestimates the energy of "peaky" signals. This means that the Chinese stimuli (less peaky) carry more energy than the German stimuli (more peaky) and could potentially be perceived as louder (although both have the same RMS!), resulting in larger TRF amplitudes. Stone and Moore (2003) also found that amplitude-compressed stimuli (less peaky) are perceived as louder compared to uncompressed stimuli (more peaky) with the same RMS. I suggest that you check this using a perceived loudness model (for instance, Zwicker implemented in Matlab), and discuss this possibility.

2. "Encoding is based on the complete TRF." You can also perform time-resolved encoding. I highly recommend doing so. This approach can likely shed light on the discrepancy between predictive power and TRF by examining the encoding process over time, which reflects the unfolding of the TRF more accurately and is thus more comparable. For methodological details, you can refer to the work of Hausfeld (2018) or Fiedler (2019).

Reviewer #2

(Remarks to the Author)

The manuscript reports on a study in which MEG was used to investigate attention-related modulations of auditory cortex activity during selective listening to native and foreign speech. To this end, groups of Chinese and German native speakers listened to Chinese and German sentences with or without competing input, and in consistent of inconsistent language combinations (i.e., C-C, G-G, G-C, C-G), and detected target words. Noteworthy, all sentences were uttered by the same (bilingual) speaker, thus minimizing the role of acoustic cues for stream segregation.

The data were analyzed using a temporal response function (TRF) approach. For single-speaker speech, differences in P1, N1, and P2 amplitudes between native vs. foreign speech are reported. When listening to competing input, an Nd wave as index of attention allocation was observed. Nd amplitude was enhanced for native speech, in particular when presented in inconsistent language combinations. The authors further investigated effects of priming on Nd wave amplitude, and compare the results of their TRF approach with a measure of encoding accuracy.

While attentional modulations in selective listening per se have been thoroughly investigated in previous studies, there is currently only little data comparing effects in native vs. foreign speech. The task design is clever, as it should allow to isolate MEG signal modulations related to "native language", largely independent of differences in low-level acoustic effects. The applied methods and analysis approaches are sound, and are generally well described.

My main concern with this manuscript is the robustness and interpretability of the shown results. The data analysis is based on relatively few trials, and the reported effects may be modulated by participant behavior/perception. As trials cannot be sorted according to participant response (due to the small number of trials), this confound is difficult to address in the statistical analysis. However, it should definitely be considered for the interpretation of the findings. Please find my detailed remarks below:

1) The TRF analysis of the two-speaker conditions are based on 30 trials each (i.e, 210 secs of data in total). Previous studies often used training sets > 10 minutes per condition, presumably resulting in a more robust TRF estimation (for a review for effects of data quantity on TRF estimation, see Mesik & Wojczak, 2022 (<https://doi.org/10.3389/fnins.2022.963629>)). Also, individual trials were rather short compared to prior studies. These factors may reduce the reliability of the measured peak amplitudes as well as the quality of the MEG signal reconstruction. This potential issue should be considered when discussing the current findings. In addition, I suggest providing measures of reliability (Cronbach's alpha) or SNR for the TRF responses/peaks.

2) As depicted in Figure 1, participants showed considerably false alarm rates across all listening conditions, reaching more than 50% in several participants. This suggests frequent failures in auditory stream segregation. As attention is often thought to act on segregated streams and not the acoustic mixture (cf. Shinn-Cunningham 2008), differences between target and masker streams may be larger than reported in the manuscript. Moreover, false alarm rates differ across conditions, so that small attention-related modulations in consistent-language trials may be attributed to failures in auditory stream segregation. To overcome this potential confound, attention-related modulations could be compared within error-free trials only. But,

given the small number of trials per condition this seems unfeasible. Therefore, the role of stream segregation for the occurrence of attentional modulations of the TRF should be considered when interpreting the results.

3) The introduction provides an overview about different factors influencing attention modulations in two-speaker paradigms. Overall, I felt that it was not optimally tailored towards the actual study. Given the scope of the manuscript on native vs. foreign language processing, I would have expected the authors to provide some literature overview about the listening advantage for native speech, and how this may affect attentional modulations. On the other hand, passages on attention switching and speaker gender, and acoustic degradation do not seem to be overly relevant in the context of the manuscript and could be removed from the introduction. Please also refer to the studies mentioned in comment #6, as these may provide additional context on native vs foreign speech processing.

4) The TRF analysis is accompanied by an encoding analysis, for better comparability with prior studies. However, if I recall the literature correctly, most of the mentioned studies used multivariate stimulus decoding approaches in which a stimulus feature (mostly, the envelope) is reconstructed from the measured MEG/EEG signals. I am not sure that the results of both analysis types can be directly compared. In broader context, I also wasn't too sure about the relevance of this analysis for understanding the native speech effect. In my view, the rationale of this second analysis needs to be explained more thoroughly, given the room it takes up in the manuscript.

5) How exactly was the time window for the Nd wave analysis determined? Based on the grand average source waveforms for target and masker streams? For the Chinese speakers, the negative deflection seems to persist beyond this time window, up to 350 – 400 msec. By contrast, the Nd wave seems to be more confined in the German speakers. Have you considered a peak-centered analysis instead of a fixed time window for the Nd wave to account for this group difference?

6) Multiple previous studies have compared speech-related activity in native vs foreign speech, partially also using natural speech and similar TRF or encoding/decoding approach (e.g., Ihara et al, 2021 (<https://doi.org/10.3389/fnhum.2021.665809>), Brodbeck et al., 2023 (<https://doi.org/10.1523/JNEUROSCI.0666-23.2023>), Di Liberto et al., 2021 (<https://doi.org/10.1016/j.neuroimage.2020.117586>), Zinsner et al., 2022 (<https://doi.org/10.1016/j.bandl.2022.105128>)). These studies seem relevant for the current experiment, as they provide further insights into the processing of native vs foreign speech, which may help explaining the observed native speech effects in the single-speaker condition.

7) As I understand the reported experimental procedures, the non-primed listening trials were presented first, followed by the primed trials. Was this counterbalanced across subjects? If not, could some of the observed behavioral differences be explained by training or time on task?

Minor comments:

- Across the manuscript, d' is reported as “stronger”. If I am not mistaken, it should read “higher”.

- The description “little experience of German” for the Chinese speakers seems vague. What was the inclusion criterion (in terms of length of residency, German class participation, etc.)? Please clarify.

- In the primed trials, were the target streams presented directly before the mixture or were there some trials in between? Please clarify the corresponding section of the methods.

- It seems that the lambda parameter was optimized using a grid search and LOOCV. Please specify the tested lambda range in the manuscript. Also, for comparison with others studies please provide the resulting lambda parameter that was used for data analysis.

- Given that the TRF was computed on the sensor level, I suggest providing a (supplementary) figure showing the TRF topography in sensor space as well as information about the dipole fit (x,y,z coordinates, direction etc.).

- Differences in envelope peak rate are linked to differences in syllable rate in the discussion. This assumption should be checked for the used stimulus material (e.g., using an online tool such as webmaus for German).

- Thank you for providing the behavioral and MEG data of the experiment in an online data repository. However, I would like to point out that the reported findings can only be reproduced by others if the stimulus material or the used envelope regressors are also shared with the neural data. I therefore suggest including these data as well.

Reviewer #3

(Remarks to the Author)

Liang, Gerwien and Gutschalk present data and results of an MEG study on listening to audiobook narrations consisting of native and/or non-native language. Participants (native Chinese and German speakers) were presented with short auditory scenes of 7s of either a single speech stream (SS) or two streams (TS) of the same speaker and were asked to press a button when they heard the target word for the relevant speaker. The authors varied target-masker language consistency (same or different language), target language (Chinese or German), priming (primed nor not primed) and acquired MEG data. Behavioral results indicated effects of each of these factors including their interactions. The authors performed an encoding analysis looking at the TRFs for source-localized activity based on the observed P1 in SS and focused on the coefficients for the delay 150-250ms for TS. The main findings were differences for target or masker stimuli, target-masker

consistency, priming and an interaction between target language and participant group. Looking at the TRF model performances, the results were different to the analysis of the TRF coefficients. The authors conclude that attention enhanced the target speech more 1) when it was in the native vs non-native language for both SS and TS and 2) when the masker stream was in a different language.

While the study has great appeal and is of clear interest to the community, in particular regarding the bilingual aspect (same narrator for both languages, large set of participants in each of the native groups), and the figures are very transparent by including single-participant points, methodological and design aspects need to be addressed before judging the results. Please find my main and minor points below.

Main points:

- TRF analysis i. The authors main results are based on the estimated TRFs that rely on an encoding analysis (via mTRFs/ridge regression). Compared to earlier studies, the stimuli are rather short (7s) which makes it difficult to get a reliable estimate for single trials. The task choice of target detection makes this even more difficult as targets (their detection and the button press) will undoubtedly contribute to the MEG signal and thus TRF estimation. To address this, one option that would not require collecting new data would be to include event predictors for the target onset in the target and masker speech as well as button press onsets (at least 3 predictors; a differentiation could be made for detected vs missed targets leading then to 5 'confound' predictors). As it is an encoding analysis, these can be included in the very same estimation in addition to the envelope derivative; the resulting TRF for the envelope derivative could then be used for further analysis (see e.g. Orb et al., 2023, iScience). This would result in responses closer to the relevant auditory activity as the other predictors would capture (most, not all) variance explained by processes unrelated to the envelope tracking. Looking at TRFs of the other predictors could then provide an ERP-like response to the coded events. As is, the TRFs and their analyses are difficult to interpret.
- TRF analysis ii. Similar to the point above, by using the encoding framework, one main advantage is that TRFs for different predictors can be estimated at once. This holds also for target and masker envelopes and including these to estimate the TRFs would remove/reduce the shared components and better delineate the TRFs for target and masker envelope derivatives.
- Priming. The motivation behind the priming manipulation remains unclear to me (I agree that it had a strong effect by making the task clearly less difficult as the same segment was repeated) and by including the interpretation of the data becomes arguably more complex. To me, the prime trials render the listening less natural as it already is (target detection of a word will be different than listening to speech; however, it constitutes a task that can be performed in known and unknown language) and I would consider the unprimed data to better reflect natural listening. Please include the motivation.
- The Nd hypothesis (l.87/88) is not integrated in the introduction and, as presented, might appear as being included after data inspection. Please introduce the Nd better or keep it more general by saying that you expect differences in the TRF. Similarly, the specification of the intervals after looking at the data requires careful analysis to avoid double-dipping. Please report which data and the statistics that was considered to define the intervals (for SS and, more importantly, TS) for readers to better judge the selection.

Minor points

- Methodological detail (TRF approach). a) I did not find information whether the first second of the stimuli was cut for the analysis. It would be more elegant to do so, as the TRF of TS will otherwise include speech without a masker material. One could argue that the analysis shouldn't start until about .5-1s after the onset of the masker stimulus as the onset of speech will affect the TRF strongly. b) It is unclear whether the authors performed permutations for the encoding/TRF. These can be used for both the TRF analysis as well as the 'encoding analysis' by providing an empirical chance level for the TRF coefficients as well as the encoding accuracy (one could argue that this is not necessary as the the comparison of TRF/encoding performance are between conditions). c) It is unclear to me whether the TRF estimation was performed on different data (P1 sources vs sensors for each hemisphere). I think it is the latter (l.297/8) and cannot come up with a reason why the analysis was redone, on different signals. An encoding analysis with mTRFs will provide both the estimated TRFs and a performance measure on how well this TRF can predict unseen data. This would make it more comparable and might already reduce the differences observed between TRF and encoding performance already. d) please provide the range of lambdas to optimize the models.
- It is unclear why reaction times were not analysed. These can provide a more sensitive, continuous outcome compared to accuracy scores with regard to task difficulty.
- Often the text about TRF results discusses '(un)attended conditions/trials', which were confusing as there are no attention conditions (participants always pay attention). Rephrasing this as 'target(masker) speech/streams' or '(un)attended speech/streams' could make this easier to follow.
- Please mention that there were no effects of hemisphere already in the beginning of the TRF analysis section (starting at l.136) to explain that the presented results are pooled across hemispheres.
- For better clarity, please provide effect size measures (for example in the ANOVA tables) and mean+/-sem for descriptive values (e.g. d-prime or TRF coefficients). For example, adding it to Table S1 and provide the same for the TS condition.
- The data set is very interesting to the community. However, I'm missing a data availability statement.

Version 1:

Reviewer comments:

Reviewer #1

(Remarks to the Author)

The authors have addressed all my previous comments thoughtfully and comprehensively. I particularly commend their effort in conducting the additional TRF analyses, which add substantial depth to the manuscript. Their careful consideration of the potential confound of loudness and the steps taken to address it are a significant strength. These revisions have greatly enhanced the clarity, rigor, and impact of the work. I am confident that this is now a strong and valuable contribution to the field.

Reviewer #2

(Remarks to the Author)

The authors have provided a thorough and comprehensive revision of their manuscript. All of my previous comments and concerns have been adequately addressed, both in their responses and within the revised manuscript. I have no further comments or suggestions.

Reviewer #3

(Remarks to the Author)

The authors addressed my concerns raised in the first review. I think that the manuscript reads more coherent and comprehensively. It is a good addition to the literature for the fields of stream segregation, bilingual processing and attention.

Dear Editor,

we wish to thank you and the reviewers for their careful reading and helpful comments. As you will see below in the point-by-point response, we have considered them carefully and improved the analysis and writing accordingly. Changes in the analysis include a revision of the behavioral response window, addition of further regressors for the TRF estimation, inclusion of Cronbach's alpha for the Nd time window, and a complete revision of the encoding analysis.

The major revision of the encoding analysis and its reporting is in response to comments from all three reviewers: our initial expectation was that the TRF analysis would linearly translate into the encoding analysis, and that the latter would therefore support our data. Then leaving this analysis out just because it did not support our analysis appeared invalid to us, but we agree that the analysis took much space in the main manuscript without contributing much to the main research question. We have now (i) tried to better explain the motivation for this analysis (ii) reduced the details in the manuscript (and moved them to the supplement), and (iii) completely revised this analysis. This includes the transformation of the analysis into source space. Moreover, we decided to perform the analysis on the complete data set in a leave-one-out fashion, instead of using only 20% of the data as in the previous analysis (where TRF and lambda values were fitted based on the other 80%). To this end, we used the fixed lambda values from the single-stream conditions (which were already used for the TRF analysis previously), such that the lambda is not individually fitted and all encoding in the leave-one-out approach can be considered "unseen". Moreover, we performed the encoding analysis separately for the P1 and Nd time windows. Nevertheless, the encoding analysis did not reproduce our main result from the TRF analysis. However, we have found strong evidence to support that this non-linear relationship is caused by an interaction of the P2 and Nd components, such that decreased Nd may cause stronger P2 in some participants, which results in better encoding accuracy of the unattended masker stream. We have included these analyses in the supplement, because we think that it is important for the field to acknowledge such non-linearities when comparing different analysis modalities.

In summary, we think that the analysis and manuscript have been significantly improved with this revision, and we hope that you and the reviewers will similarly appreciate this revised version of our manuscript.

Yours sincerely, Alexander Gutschalk, Meng Liang, and Johannes Gerwien

reviewer #1

Brief summary of the manuscript:

The author asserts that while the advantage of native speech comprehension is widely acknowledged, the underlying neural mechanisms behind this phenomenon remain elusive. To address this gap, the authors conducted a study employing magnetoencephalography (MEG) to investigate whether attentional enhancement in the auditory cortex is more pronounced for native speech. They utilized bilingual speakers producing Chinese and German speech, creating a simulated cocktail party scenario with both consistent and inconsistent language combinations. Native speakers of Chinese and German participated in a target detection task within the designated speech stream.

The findings revealed that attentional enhancement manifested as increased negative activity in the temporal response function. Notably, this enhancement was more pronounced when the target speech was in the participants' native language compared to a non-native language. Furthermore, it was heightened when presented with inconsistent language combinations compared to consistent ones. The authors suggest that this heightened neural activity for native speech may be attributed to a more accurate top-down prediction mechanism specifically tuned to the native speech stream.

In summary, the study sheds light on the neural underpinnings of the listening advantage for native speech, highlighting the role of attentional mechanisms in facilitating comprehension, particularly in challenging linguistic environments.

Overall impression of the work:

Overall, this manuscript presents a well-executed study investigating the neural basis of the listening advantage for native speech comprehension. The writing is clear and concise, effectively conveying the research aims, methods, and results. The figures provided are visually appealing and aid in understanding the experimental setup and findings.

The study design, incorporating magnetoencephalography (MEG) to examine attentional enhancement in the auditory cortex, is methodologically sound. By utilizing bilingual speakers and creating a realistic cocktail party scenario with both consistent and inconsistent language combinations, the authors were able to simulate real-world listening conditions and capture relevant neural responses.

The results of the study provide valuable insights into the mechanisms underlying native language processing, demonstrating heightened attentional enhancement for native speech compared to non-native speech. Additionally, the finding that attentional enhancement was more pronounced in response to inconsistent language combinations adds nuance to our understanding of language

processing in complex environments.

Overall, this manuscript contributes significantly to the literature on language comprehension and attentional mechanisms. The thoroughness of the study design, clarity of presentation, and compelling findings make it a noteworthy addition to the field. While the manuscript is strong, a few additional analyses could further strengthen the work (see comments above).

Specific comments, with recommendations for addressing each comment

	Questions	reply
1	36: Listening to native speech at usual levels...487: Have their hearing thresholds been measured?	Participants did not have formal audiometry, but they were young and did not have any medical history of hearing impairment (added in line 519 “ All participants reported normal hearing.”), which is why we consider it very likely that audiometry would have been within normal limits. Moreover, the SPL was 80dB for mixtures (added in line 549 “Stimuli were presented diotically with an average sound pressure level of 80 dB for the combined speech streams, measured with a Brüel & Kjaer microphone (Ear Simulator Type 4157) in a 2cc coupler.”), which is a comfortable level but also well above threshold.
2	516: Stimuli were presented diotic?	Yes. Added on line 549. “Stimuli were presented diotically with an average sound pressure level of 80 dB for the combined speech streams, measured with a Brüel & Kjaer microphone (Ear Simulator Type 4157) in a 2cc coupler.”

3	525: Why a 2-s time window? I would suggest a time window that exclude biological implausible early responses for instance (150ms-2000ms, see Bendixen, 2013). A histogram of hits and false alarms could be helpful to set the boundaries of the window.	The time interval has been modified from 2 s after target onset to 2 s after target offset (i.e. the name "Jack").
4	False alarms were solely related to button presses outside the window of the target word? Maybe it would be worth checking if some of the false alarms are appeared out of nowhere or if some of the false alarms are related to the target word in the masker stream.	Most of the false alarms occurred in the context of the target name in the masker stream. We therefore modified the analysis such that only responses to the masker stream were considered false alarms (added on line 560 “Button presses in response to the target word presented within the masker stream were considered false alarms.”). (Note that this change resulted in a change for the one-stream condition: because of the low FA rate, many subjects had been on ceiling here. Since there is no second stream in this condition, we now changed to hit rate instead of d' for this case. The results now show a native-language effect for this condition as well, which was not reported in the original submission where the analysis was based on d').
5	97: Was the same target stream presented twice? If so, could be habituation a potential confound?	Each target stream was only used in one trial over the whole experiment (cf. methods line 546 “Each mixture was used only once in the whole experiment.”). The only repetition was from the single-stream prime to the two-stream mixture.

6	99-105: Figure 1: would add also the explanation of the two different bars (primed/unprimed) to figure caption	An explanation of the two bar types has been added in the caption of Figure 1.
7	148: would help to indicate significant windows already in the plot	We have considered this suggestion and then decided not to indicate significance of the single conditions, because they provide only limited information regarding the interactions and across group effects, which are more important. Also, indicating the significance will vary quite a bit depending on the level of correction for multiple comparison. We instead present the raw data with standard errors, the distance between which provides an indirect estimate of significant differences in Figure 1 as well as in the other figures.
8	307: different citation style.	done
9	381: un-primed unprimed	done

10	Could the higher TRF amplitudes of the P1 and P2 components also be driven by a difference in perceived loudness between Chinese and German stimuli? Due to its mathematical nature (squaring), the RMS underestimates the energy of "peaky" signals. This means that the Chinese stimuli (less peaky) carry more energy than the German stimuli (more peaky) and could potentially be perceived as louder (although both have the same RMS!), resulting in larger TRF amplitudes. Stone and Moore (2003) also found that amplitude-compressed stimuli (less peaky) are perceived as louder compared to uncompressed stimuli (more peaky) with the same RMS. I suggest that you check this using a perceived loudness model (for instance, Zwicker implemented in Matlab), and discuss this possibility.	We calculated the model-based loudness as suggested by the reviewer. The results show a small difference, but one that rather suggests a somewhat stronger loudness for the German stimuli. Accordingly, the P1 difference cannot be explained based on loudness. Results of this analysis have been included in lines 193 - 198 in the results section and implication are discussed on lines 370 – 372 in the discussion section “The alternative explanation that a higher perceived loudness of the Chinese stimuli contributes to the effect, which may be due to language-specific differences, was not supported by a computational loudness model.”.
----	--	---

11	"Encoding is based on the complete TRF." You can also perform time-resolved encoding. I highly recommend doing so. This approach can likely shed light on the discrepancy between predictive power and TRF by examining the encoding process over time, which reflects the unfolding of the TRF more accurately and is thus more comparable. For methodological details, you can refer to the work of Hausfeld (2018) or Fiedler (2019).	The encoding analysis has been completely revised. We have also added a time-resolved variant of the encoding to the supplement (Supplementary Fig. 5, 6). The time resolved analysis suggests that the P1 time window dominates the encoding in the complete time window. Moreover, we have added a subset analysis of an exemplary condition, which explains why the Nd effect does not linearly translate into the encoding analysis (Supplementary Fig. 7).
----	--	--

reviewer #2

The manuscript reports on a study in which MEG was used to investigate attention-related modulations of auditory cortex activity during selective listening to native and foreign speech. To this end, groups of Chinese and German native speakers listened to Chinese and German sentences with or without competing input, and in consistent of inconsistent language combinations (i.e., C-C, G-G, G-C, C-G), and detected target words. Noteworthy, all sentences were uttered by the same (bilingual) speaker, thus minimizing the role of acoustic cues for stream segregation.

The data were analyzed using a temporal response function (TRF) approach. For single-speaker speech, differences in P1, N1, and P2 amplitudes between native vs. foreign speech are reported. When listening to competing input, an Nd wave as index of attention

allocation was observed. Nd amplitude was enhanced for native speech, in particular when presented in inconsistent language combinations. The authors further investigated effects of priming on Nd wave amplitude, and compare the results of their TRF approach with a measure of encoding accuracy.

While attentional modulations in selective listening per se have been thoroughly investigated in previous studies, there is currently only little data comparing effects in native vs. foreign speech. The task design is clever, as it should allow to isolate MEG signal modulations related to “native language”, largely independent of differences in low-level acoustic effects. The applied methods and analysis approaches are sound, and are generally well described.

My main concern with this manuscript is the robustness and interpretability of the shown results. The data analysis is based on relatively few trials, and the reported effects may be modulated by participant behavior/perception. As trials cannot be sorted according to participant response (due to the small number of trials), this confound is difficult to address in the statistical analysis. However, it should definitely be considered for the interpretation of the findings. Please find my detailed remarks below:

	Questions	reply
1	The TRF analysis of the two-speaker conditions are based on 30 trials each (i.e, 210 secs of data in total). Previous studies often used training sets > 10 minutes per condition, presumably resulting in a more robust TRF estimation (for a review for effects of data quantity on TRF estimation, see Mesik & Wojczak, 2022 (https://doi.org/10.3389/fnins.2022.963629)). Also, individual trials were rather short compared to prior studies. These factors may reduce the reliability of the measured peak amplitudes as well as the quality of the MEG signal reconstruction. This potential issue should be considered when discussing the current findings. In addition, I suggest providing measures of reliability (Cronbach’s alpha) or SNR for the TRF responses/peaks.	We have added an evaluation of the TRF based on Cronbach's alpha, which shows values around 0.9 for the TRF source waveforms in the Nd time windows (line 215 - 217 “An analysis of data reliability in the Nd time interval, based on Cronbach's alpha calculated across participants and all 16 conditions, revealed a satisfactory values of 0.953 for Chinese 0.918 for German listeners.”). The encoding analysis was changed to a leave-one-out process to enhance the amount of data evaluated. Finally, we discuss what implications the relatively short time intervals per condition may have, in particular for the encoding analysis (TRF versus MEG encoding; line 448 - 474).

2	As depicted in Figure 1, participants showed considerably false alarm rates across all listening conditions, reaching more than 50% in several participants. This suggests frequent failures in auditory stream segregation. As attention is often thought to act on segregated streams and not the acoustic mixture (cf. Shinn-Cunningham 2008), differences between target and masker streams may be larger than reported in the manuscript. Moreover, false alarm rates differ across conditions, so that small attention-related modulations in consistent-language trials may be attributed to failures in auditory stream segregation. To overcome this potential confound, attention-related modulations could be compared within error-free trials only. But, given the small number of trials per condition this seems unfeasible. Therefore, the role of stream segregation for the occurrence of attentional modulations of the TRF should be considered when interpreting the results.	We agree with R2 that the performance in the consistent-language trials - with d' not significantly different from zero - suggests that the two streams cannot be segregated or alternatively are confused as the stimulus proceeds. In this case, even limiting the analysis to error free trials (which are indeed too few for a subanalysis) may produce the same result, because hit and miss trials are overall at chance level, and a lack of a FA could as well be a good guess. To acknowledge these limitations and discuss different options for the interpretation of the present findings, we have added a section in the discussion of stream segregation in the context of the priming paradigm (line 419 - 447).
---	--	---

3	The introduction provides an overview about different factors influencing attention modulations in in two-speaker paradigms. Overall, I felt that it was not optimally tailored towards the actual study. Given the scope of the manuscript on native vs. foreign language processing, I would have expected the authors to provide some literature overview about the listening advantage for native speech, and how this may affect attentional modulations. On the other hand, passages on attention switching and speaker gender, and acoustic degradation do not seem to be overly relevant in the context of the manuscript and could be removed from the introduction. Please also refer to the studies mentioned in comment #6, as these may provide additional context on native vs foreign speech processing.	A limitation is that most of the literature on non-native speech is on L2 speech, i.e. speech that participants can interpret. While we think that our data are also relevant for L2-speech perception under masking, many of the L2 results cannot be used to make predictions for our data, because participants in our study did not understand the speech material. We have tried to make the difference explicit by using (and defining) the term “foreign” speech in the introduction (second paragraph, line 72 - 74 “The comparison of native with foreign speech (i.e. speech in a language that is unknown to the participants), both masked by noise, revealed a trend for higher neural activity and better reconstruction of MEG signals for the foreign speech.”). In response to the reviewer, we have nevertheless extended the part on non-native speech perception under masking and removed the sentence on attention switching instead. Further reference to the L2 literature recommended by the reviewer is taken up in the discussion. We have kept the part on degraded speech, however, because it is similar to foreign speech in so far as it cannot (in the extreme form) be understood. There is an ongoing discussion whether the understandability of degraded speech has an influence on the auditory cortex response related to the envelope. Therefore, predictions based on degraded speech are relevant for the introduction of the experiment, in our view.
---	---	---

4	The TRF analysis is accompanied by an encoding analysis, for better comparability with prior studies. However, if I recall the literature correctly, most of the mentioned studies used multivariate stimulus decoding approaches in which a stimulus feature (mostly, the envelope) is reconstructed from the measured MEG/EEG signals. I am not sure that the results of both analysis types can be directly compared. In broader context, I also wasn't too sure about the relevance of this analysis for understanding the native speech effect. In my view, the rationale of this second analysis needs to be explained more thoroughly, given the room it takes up in the manuscript.	The encoding was not part of our primary hypothesis and we agree with the reviewer that this analysis is not relevant for our conclusions in the end. We initially expected that this analysis would confirm our primary results and thought it would be good to include, as many recent papers report encoding or decoding rather than TRFs. The reviewer is also correct that the studies we directly refer to used decoding rather than encoding, but in our case the two techniques provided very consistent results. Considering comments of reviewers 1 & 3, we have substantially revised the encoding analysis. After all, we think that we can explain relatively well why the TRF and encoding analysis are not linearly related (which we think is of general interest for future comparison between these analysis approaches). In response to Reviewer 2's comment, we have substantially reduced the space used in the main manuscript for reporting the encoding analysis. Most of the additional, exploratory analyses have instead been moved to the supplement. Moreover, we have omitted the parallel report of the decoding analysis, because it appeared redundant and would have taken even more space (note that the similarity of the analysis holds for the new analysis in the revised paper, but was omitted to simplify the presentation).
---	---	---

5	How exactly was the time window for the Nd wave analysis determined? Based on the grand average source waveforms for target and masker streams? For the Chinese speakers, the negative deflection seems to persist beyond this time window, up to 350 – 400 msec. By contrast, the Nd wave seems to be more confined in the German speakers. Have you considered a peak-centered analysis instead of a fixed time window for the Nd wave to account for this group difference?	The Nd time window was defined based on the grand-average TRF of all target trials, starting after the P1 and including the rise of the negative-going response including the first superimposed peak in the grand average after 210 ms. A detailed description has been added in the result section (line 212 – 214 “For the analysis of attention effects, the Nd was defined based on the grand average across all target stream conditions (Supplemental Fig. 4).”), and the grand average used for the definition has been added in Supplementary Fig. 4 and details in the methods section (line 640 “A 100-ms-long time interval from 150 - 250 ms was therefore chosen, starting after the P1 and covering the rising part of the negativity including the first superimposed peak in the grand average.”). We have decided to use a fixed analysis window instead of an individual, peak-based analysis, because the latter is problematic for several reasons: first, not all conditions exhibit a clear N1/Nd response, such that using a peak analysis sometimes measures arbitrary peaks within the noise or returns measures at the end of the time window. Second, the comparison of attended target and unattended masker conditions would typically result in different latencies, which makes a comparison of the two difficult. Generally, the attentional modulation in auditory cortex, i.e. N1 or Nd, has been described as transient after P1, which is why we did not extend the time interval further to also cover the full sustained response. We agree with the reviewer that the separation of where the transient interval ends and the sustained interval starts is difficult due to ambiguous indicators for boundaries.
---	--	--

6	Multiple previous studies have compared speech-related activity in native vs foreign speech, partially also using natural speech and similar TRF or encoding/decoding approach (e.g., Ihara et al, 2021 (https://doi.org/10.3389/fnhum.2021.665809), Brodbeck et al., 2023(https://doi.org/10.1523/JNEUROSCI.0666-23.2023), Di Liberto et al., 2021 (https://doi.org/10.1016/j.neuroimage.2020.117586), Zinsner et al., 2022 (https://doi.org/10.1016/j.bandl.2022.105128)). These studies seem relevant for the current experiment, as they provide further insights into the processing of native vs foreign speech, which may help explaining the observed native speech effects in the single-speaker condition.	We have added three of the studies to the discussion (line 486, line 491, and line 496). We found it difficult to relate the study by Ihara et al to our data, first, because the analysis is mainly on speech proficiency and, second, because the analysis is based on word onset, both of which cannot be directly compared to our data. Note however that these studies are also on L1 versus L2 and not on L1 versus foreign speech (cf. reply to comment 3).
7	As I understand the reported experimental procedures, the non-primed listening trials were presented first, followed by the primed trials. Was this counterbalanced across subjects? If not, could some of the observed behavioral differences be explained by training or time on task?	This is correct, the primed stimuli were always presented in the second half of the recording session. Thus, we cannot exclude any training effects that could additionally contribute to the difference between primed and unprimed conditions. On the other hand, since a general difference in Nd amplitude was not observed but only an enhanced negative-going response for target and masker streams alike, we consider the possible influence of practice less likely. We added a sentence in the results section (line 272 “Thus, an additional effect of practice in addition to priming cannot be excluded.”) and one in the discussion (line 446 “Finally, since unprimed blocks always preceded primed blocks in the current

		study, a practice effect can also not be ruled out completely.”) to point out the possibility of practice effects.
8	Across the manuscript, d' is reported as “stronger”. If I am not mistaken, It should read “higher”.	Thank you for spotting, this has been corrected.
9	The description “little experience of German“ for the Chinese speakers seems vague. What was the inclusion criterion (in terms of length of residency, German class participation, etc.)? Please clarify.	Most of the participants had done a four-weeks German class before moving to Heidelberg, but they did not report significant understanding of German. This and the duration that participants had been living in Germany has been added to the methods section (line 516 – 519 “13 of the Chinese participants had participated in a one-month German course before moving to Germany, but none of them reported to understand German when they participated in the experiment. The average time that Chinese participants had spent in Germany was 8.5 months (3 - 15months).”).
10	In the primed trials, were the target streams presented directly before the mixture or were there some trials in between? Please clarify the corresponding section of the methods.	Primes were presented directly before the corresponding mixtures. This information has been added in the beginning of the results section (line 107 “In the second half of the experiment, the primed condition was presented, where the complete target stream was presented alone directly before the presentation of the mixture.”) and in the methods

		section (line 545 “In the second section of the experiment, the target streams were presented once as prime before the mixtures.”).
11	It seems that the lambda parameter was optimized using a grid search and LOOCV. Please specify the tested lambda range in the manuscript. Also, for comparison with others studies please provide the resulting lambda parameter that was used for data analysis.	The range for search and the resulting lambda values have been added in the methods section (line 594 – 596 “The optimal lambda parameter was searched for in the range from 10^{-6} to 10^{10} , yielding values in a range from 10^3 to 10^6 (6×10^3 , 27×10^5 , 1×10^6).”).
12	Given that the TRF was computed on the sensor level, I suggest providing a (supplementary) figure showing the TRF topography in sensor space as well as information about the dipole fit (x,y,z coordinates, direction etc.).	A figure that maps the TRF topography for prime, target, and masker TRFs has been added as Supplementary Fig. 3. We also added the dipole coordinates in the new Supplementary Table 1 and also show them in the new Figure panel e of Fig. 2 in the main manuscript.
13	Differences in envelope peak rate are linked to differences in syllable rate in the discussion. This assumption should be checked for the used stimulus material (e.g., using an online tool such as webmaus for German).	the syllable rate has been measured and added to the manuscript for comparison (results, line 190 – 191 “The syllable rate amounted to 5.4/s in German and 4.2 in Chinese, confirming that envelope peak rate and syllable rate are proportional to each other.”).

14	Thank you for providing the behavioral and MEG data of the experiment in an online data repository. However, I would like to point out that the reported findings can only be reproduced by others if the stimulus material or the used envelope regressors are also shared with the neural data. I therefore suggest including these data as well.	The data have now been uploaded to a repository including the regressors required for their evaluation under a private URL: https://heidata.uni-heidelberg.de/privateurl.xhtml?token=0ad0078f-3e32-470d-8b2b-a357b5eb6df3
----	---	--

reviewer #3

Liang, Gerwien and Gutschalk present data and results of an MEG study on listening to audiobook narrations consisting of native and/or non-native language. Participants (native Chinese and German speakers) were presented with short auditory scenes of 7s of either a single speech stream (SS) or two streams (TS) of the same speaker and were asked to press a button when they heard the target word for the relevant speaker. The authors varied target-masker language consistency (same or different language), target language (Chinese or German), priming (primed nor not primed) and acquired MEG data. Behavioral results indicated effects of each of these factors including their interactions. The authors performed an encoding analysis looking at the TRFs for source-localized activity based on the observed P1 in SS and focused on the coefficients for the delay 150-250ms for TS. The main findings were differences for target or masker stimuli, target-masker consistency, priming and an interaction between target language and participant group. Looking at the TRF model performances, the results were different to the analysis of the TRF coefficients. The authors conclude that attention enhanced the target speech more 1) when it was in the native vs non-native language for both SS and TS and 2) when the masker stream was in a different language.

While the study has great appeal and is of clear interest to the community, in particular regarding the bilingual aspect (same narrator for both languages, large set of participants in each of the native groups), and the figures are very transparent by including single-participant points, methodological and design aspects need to be addressed before judging the results. Please find my main and minor points below.

	Questions	reply
1	TRF analysis i. The authors main results are based on the estimated TRFs that rely on an encoding analysis (via mTRFs/ridge regression). Compared to earlier studies, the stimuli are rather short (7s) which makes it difficult to get a reliable estimate for single trials. The task choice of target detection makes this even more difficult as targets (their detection and the button press) will undoubtedly contribute to the MEG signal and thus TRF estimation. To address this, one option that would not require collecting new data would be to include event predictors for the target onset in the target and masker speech as well as button press onsets (at least 3 predictors; a differentiation could be made for detected vs missed targets leading then to 5 ‘confound’ predictors). As it is an encoding analysis, these can be included in the very same estimation in addition to the envelope derivative; the resulting TRF for the envelope derivative could then be used for further analysis (see e.g. Orb et al., 2023, iScience). This would result in responses closer to the relevant auditory activity as the other predictors would capture (most, not all) variance explained by processes unrelated to the envelope tracking. Looking at TRFs of the other predictors could then provide an ERP-like response to the coded events. As is, the TRFs and their analyses are difficult to interpret.	The data have been re-analyzed as suggested (i.e. using the envelope from the previous analysis, the onset of the target word, the onset of the same word in the distractor, and the response as fifths event), modeling target, distractor "Jack", and responses as additional events. The results are presented in the revised paper instead of the previous analysis.

2	TRF analysis ii. Similar to the point above, by using the encoding framework, one main advantage is that TRFs for different predictors can be estimated at once. This holds also for target and masker envelopes and including these to estimate the TRFs would remove/reduce the shared components and better delineate the TRFs for target and masker envelope derivatives.	The target and masker envelope had already been estimated together in the initial version of the analysis, this was maybe not explained well enough, the methods description of the encoding analysis has been revised. (Note that we did not include the additional regressors for the encoding analysis, because the latter is now focused on the auditory cortex and the additional components are mostly generated elsewhere).
3	Priming. The motivation behind the priming manipulation remains unclear to me (I agree that it had a strong effect by making the task clearly less difficult as the same segment was repeated) and by including the interpretation of the data becomes arguably more complex. To me, the prime trials render the listening less natural as it already is (target detection of a word will be different than listening to speech; however, it constitutes a task that can be performed in known and unknown language) and I would consider the unprimed data to better reflect natural listening. Please include the motivation.	The priming task was motivated for comparability with the (as far as we know) only MEG/EEG study that had used the same speaker in a mixture before (Wang et al. 2019). Since that study had suggested that an attention effect was only observed with priming, we had included this condition to avoid seeing no effects at all. This information has been added in the motivation at the end of the introduction (line 97 “As a previous study with a same-speaker setup had found an attention effect only when the target stream was primed with the isolated target stream, we added a comparable condition to our setup.”).

4	The Nd hypothesis (l.87/88) is not integrated in the introduction and, as presented, might appear as being included after data inspection. Please introduce the Nd better or keep it more general by saying that you expect differences in the TRF. Similarly, the specification of the intervals after looking at the data requires careful analysis to avoid double-dipping. Please report which data and the statistics that was considered to define the intervals (for SS and, more importantly, TS) for readers to better judge the selection.	We now introduce the Nd concept in the revised introduction (line 60 “In the context of speech streams, this negative-going wave has mostly been referred to as N1/N100 whereas studies investigating attention to tone streams have variably referred to it as N1, processing negativity, or neutrally to the negative difference wave (Nd)”). The observation that attentional modulation in auditory cortex is mostly observed by negative-going responses is quite accepted in the literature using tone streams, whereas the nomenclature has been somewhat different for the attention to speech streams. Another motivation for using Nd rather than N1 here is our own work, where we previously found negative-going activity for attended streams (referred to N1 by that time) that overlapped in latency with the P2, the latter of which was mostly observed in unattended trials, only. The time intervals for the statistical analysis were defined based on the grand average for the primes and the grand average for the target streams, such that their definition should have no influence on the comparisons that are of interest for this study (in the case of Nd it could be argued that the overall attention effect was biased by this choice. However, this does not apply to the interactions with the attention effect, which are the research question of this study). The procedure is now described in more detail in the methods section (line 640 “A 100-ms-long time interval from 150 - 250 ms was therefore chosen, starting after the P1 and covering the rising part of the negativity including the first superimposed peak in the grand average.”), referenced in the
---	---	---

		results section (line 212 – 214 “For the analysis of attention effects, the Nd time window was defined based on the grand average across all target stream conditions (Supplemental Fig. 4).”), and shown in the new supplementary Fig. 4. During the literature review for this revision, we realized that a number of papers also report enhanced P2 for attended speech streams, but did not find any discussion/explanation of/for why some studies find this and other (like ours) do not. We have therefore referenced this point in the discussion (in the context of the encoding analysis, for which it is particularly relevant; see line 450 - 460).
--	--	---

5	Methodological detail (TRF approach). a) I did not find information whether the first second of the stimuli was cut for the analysis. It would be more elegant to do so, as the TRF of TS will otherwise include speech without a masker material. One could argue that the analysis shouldn't start until about .5-1s after the onset of the masker stimulus as the onset of speech will affect the TRF strongly. b) It is unclear whether the authors performed permutations for the encoding/TRF. These can be used for both the TRF analysis as well as the 'encoding analysis' by providing an empirical chance level for the TRF coefficients as well as the encoding accuracy (one could argue that this is not necessary as the the comparison of TRF/encoding performance are between conditions). c) It is unclear to me whether the TRF estimation was performed on different data (P1 sources vs sensors for each hemisphere). I think it is the latter (l.297/8) and cannot come up with a reason why the analysis was redone, on different signals. An encoding analysis with mTRFs will provide both the estimated TRFs and a performance measure on how well this TRF can predict unseen data. This would make it more comparable and might already reduce the differences observed between TRF and encoding performance already. d) please provide the range of lambdas to optimize the models.	(a) the first second was cut from the streams, more details have been added in methods (line 579 "For the analysis of the mixtures, the cues - i.e. the first second of the target stimuli - were not included in the analysis."). In the process of this revision, we have also tested a version where further 0.5 s at the beginning of the second stream were omitted. While this analysis produced generally the same results in the statistical analysis in the Nd time window (the effects were even somewhat stronger), the waveforms coming out of this analysis were more noisy, probably because of the further reduction of stimulus material. We therefore decided to stick with the time intervals that include the onset of the masker stream. (b) We did not perform a permutation test. In our understanding this is not required for the comparison performed here. (c) We have now found a way to transform the raw data into source waveforms and have changed the encoding analysis to be in source space, as well. The results are rather similar to the previous, sensor-based analysis, and did not resolve the differences between the TRF and encoding results. (d) Lambda values have been added to the methods section (line 594 – 596 "The optimal lambda parameter was searched for in the range from 10⁻⁶ to 10¹⁰, yielding values in a range from 10³ to 10⁶ (6x 10³, 27x 10⁵, 1x10⁶).").
---	---	---

6	It is unclear why reaction times were not analysed. These can provide a more sensitive, continuous outcome compared to accuracy scores with regard to task difficulty.	Reaction times have now been evaluated and produced overall similar results as the d' analysis (except for the comparison between primed and unprimed trials). These results are now briefly mentioned in the results section and are shown in supplementary Fig. 1 and 2.
7	Often the text about TRF results discusses '(un)attended conditions/trials', which were confusing as there are no attention conditions (participants always pay attention). Rephrasing this as 'target(masker) speech/streams' or '(un)attended speech/streams' could make this easier to follow.	The text has been edited as suggested.
8	Please mention that there were no effects of hemisphere already in the beginning of the TRF analysis section (starting at l.136) to explain that the presented results are pooled across hemispheres.	This information has been added in line 164 - 166("Since no hemisphere main effects or interactions with hemisphere were observed, the average of left- and right hemisphere activity is reported below.") and in line 217 – 220("For the attention effect in the Nd time interval, no significant main effects of or first-order interactions with hemisphere were observed, such that the statistics below do not report any hemisphere effects for clarity.").
9	For better clarity, please provide effect size measures (for example in the ANOVA tables) and mean+/-sem for descriptive values (e.g. d-prime or TRF coefficients). For example, adding it to Table S1 and provide the same for the TS condition.	Effect-size values (partial eta square) have been added to all tables. Mean and sem are provided in the figures for the single conditions.

10	The data set is very interesting to the community. However, I'm missing a data availability statement.	The data are available online for review and will be published in final version together with the paper.
----	--	--

a Chinese Native Listeners

c

e

b German Native Listeners

d

f

updates:

1. reponses time window changed from 2s time window from onset of the target word to offset of the target word
2. Only the button presses that appeared in the 2s time window after offset of the target word from unattended streams were considered as false alarms.

updates:

1. Average dipole positions fitted to the P1 projected on a standard brain were added.
2. Hit rate for speech in quiet was included in this figure.
3. In addition to envelope, two more predictors (target word position and button presses) were given to the regression.

updates:

1. Five predictors(attended speech envelope, unattended speech envelop, target word from attended speech, target word from unattended speech and button presses) was included.

updates:

1. Five predictors (attended speech envelope, unattended speech envelope, target word from attended speech, target word from unattended speech and button presses) was included.

updates:

1. The prediction accuracy shown here was the mean from leave-one-out cross validation(all data) instead of prediction accuracy of unseen data.
2. The lambda was the optimal lambda fitted to envelope of speech in quiet.

Dear Reviewers,

we wish to thank you for your careful reading and helpful comments. As you will see below in the point-by-point response, we have considered them carefully and improved the analysis and writing accordingly. Changes in the analysis include a revision of the behavioral response window, addition of further regressors for the TRF estimation, inclusion of Cronbach's alpha for the Nd time window, and a complete revision of the encoding analysis.

The major revision of the encoding analysis and its reporting is in response to comments from all three reviews: our initial expectation was that the TRF analysis would linearly translate into the encoding analysis, and that the latter would therefore support our data. Then leaving this analysis out just because it did not support our analysis appeared invalid to us, but we agree that the analysis took much space in the main manuscript without contributing much to the main research question. We have now (i) tried to better explain the motivation for this analysis (ii) reduced the details in the manuscript (and moved them to the supplement), and (iii) completely revised this analysis. This includes the transformation of the analysis into source space. Moreover, we decided to perform the analysis on the complete data set in a leave-one-out fashion, instead of using only 20% of the data as in the previous analysis (where TRF and lambda values were fitted based on the other 80%). To this end, we used the fixed lambda values from the single-stream conditions (which were already used for the TRF analysis previously), such that the lambda is not individually fitted and all encoding in the leave-one-out approach can be considered "unseen". Moreover, we performed the encoding analysis separately for the P1 and Nd time windows. Nevertheless, the encoding analysis did not reproduce our main result from the TRF analysis. However, we have found strong evidence to support that this non-linear relationship is caused by an interaction of the P2 and Nd components, such that decreased Nd may cause stronger P2 in some participants, which results in better encoding accuracy of the unattended masker stream. We have included these analyses in the supplement, because we think that it is important for the field to acknowledge such non-linearities when comparing different analysis modalities.

In summary, we think that the analysis and manuscript have been significantly improved with this revision, and we hope that you will similarly appreciate this revised version of our manuscript.

Yours sincerely, Alexander Gutschalk, Meng Liang, and Johannes Gerwien

reviewer #1

Brief summary of the manuscript:

The author asserts that while the advantage of native speech comprehension is widely acknowledged, the underlying neural mechanisms behind this phenomenon remain elusive. To address this gap, the authors conducted a study employing magnetoencephalography (MEG) to investigate whether attentional enhancement in the auditory cortex is more pronounced for native speech. They utilized bilingual speakers producing Chinese and German speech, creating a simulated cocktail party scenario with both consistent and inconsistent language combinations. Native speakers of Chinese and German participated in a target detection task within the designated speech stream.

The findings revealed that attentional enhancement manifested as increased negative activity in the temporal response function. Notably, this enhancement was more pronounced when the target speech was in the participants' native language compared to a non-native language. Furthermore, it was heightened when presented with inconsistent language combinations compared to consistent ones. The authors suggest that this heightened neural activity for native speech may be attributed to a more accurate top-down prediction mechanism specifically tuned to the native speech stream.

In summary, the study sheds light on the neural underpinnings of the listening advantage for native speech, highlighting the role of attentional mechanisms in facilitating comprehension, particularly in challenging linguistic environments.

Overall impression of the work:

Overall, this manuscript presents a well-executed study investigating the neural basis of the listening advantage for native speech comprehension. The writing is clear and concise, effectively conveying the research aims, methods, and results. The figures provided are visually appealing and aid in understanding the experimental setup and findings.

The study design, incorporating magnetoencephalography (MEG) to examine attentional enhancement in the auditory cortex, is methodologically sound. By utilizing bilingual speakers and creating a realistic cocktail party scenario with both consistent and inconsistent language combinations, the authors were able to simulate real-world listening conditions and capture relevant neural responses.

The results of the study provide valuable insights into the mechanisms underlying native language processing, demonstrating

heightened attentional enhancement for native speech compared to non-native speech. Additionally, the finding that attentional enhancement was more pronounced in response to inconsistent language combinations adds nuance to our understanding of language processing in complex environments.

Overall, this manuscript contributes significantly to the literature on language comprehension and attentional mechanisms. The thoroughness of the study design, clarity of presentation, and compelling findings make it a noteworthy addition to the field. While the manuscript is strong, a few additional analyses could further strengthen the work (see comments above).

Specific comments, with recommendations for addressing each comment

	Questions	reply
1	36: Listening to native speech at usual levels...487: Have their hearing thresholds been measured?	Participants did not have formal audiometry, but they were young and did not have any medical history of hearing impairment (added in line 519 “ All participants reported normal hearing.”), which is why we consider it very likely that audiometry would have been within normal limits. Moreover, the SPL was 80dB for mixtures (added in line 549 “Stimuli were presented diotically with an average sound pressure level of 80 dB for the combined speech streams, measured with a Brüel & Kjaer microphone (Ear Simulator Type 4157) in a 2cc coupler.”), which is a comfortable level but also well above threshold.
2	516: Stimuli were presented diotic?	Yes. Added on line 549. “Stimuli were presented diotically with an average sound pressure level of 80 dB for the combined speech streams, measured with a Brüel & Kjaer microphone (Ear Simulator Type 4157) in a 2cc coupler.”

3	525: Why a 2-s time window? I would suggest a time window that exclude biological implausible early responses for instance (150ms-2000ms, see Bendixen, 2013). A histogram of hits and false alarms could be helpful to set the boundaries of the window.	The time interval has been modified from 2 s after target onset to 2 s after target offset (i.e. the name "Jack").
4	False alarms were solely related to button presses outside the window of the target word? Maybe it would be worth checking if some of the false alarms are appeared out of nowhere or if some of the false alarms are related to the target word in the masker stream.	Most of the false alarms occurred in the context of the target name in the masker stream. We therefore modified the analysis such that only responses to the masker stream were considered false alarms (added on line 560 “Button presses in response to the target word presented within the masker stream were considered false alarms.”). (Note that this change resulted in a change for the one-stream condition: because of the low FA rate, many subjects had been on ceiling here. Since there is no second stream in this condition, we now changed to hit rate instead of d' for this case. The results now show a native-language effect for this condition as well, which was not reported in the original submission where the analysis was based on d').
5	97: Was the same target stream presented twice? If so, could be habituation a potential confound?	Each target stream was only used in one trial over the whole experiment (cf. methods line 546 “Each mixture was used only once in the whole experiment.”). The only repetition was from the single-stream prime to the two-stream mixture.

6	99-105: Figure 1: would add also the explanation of the two different bars (primed/unprimed) to figure caption	An explanation of the two bar types has been added in the caption of Figure 1.
7	148: would help to indicate significant windows already in the plot	We have considered this suggestion and then decided not to indicate significance of the single conditions, because they provide only limited information regarding the interactions and across group effects, which are more important. Also, indicating the significance will vary quite a bit depending on the level of correction for multiple comparison. We instead present the raw data with standard errors, the distance between which provides an indirect estimate of significant differences in Figure 1 as well as in the other figures.
8	307: different citation style.	done
9	381: un-primed unprimed	done

10	Could the higher TRF amplitudes of the P1 and P2 components also be driven by a difference in perceived loudness between Chinese and German stimuli? Due to its mathematical nature (squaring), the RMS underestimates the energy of "peaky" signals. This means that the Chinese stimuli (less peaky) carry more energy than the German stimuli (more peaky) and could potentially be perceived as louder (although both have the same RMS!), resulting in larger TRF amplitudes. Stone and Moore (2003) also found that amplitude-compressed stimuli (less peaky) are perceived as louder compared to uncompressed stimuli (more peaky) with the same RMS. I suggest that you check this using a perceived loudness model (for instance, Zwicker implemented in Matlab), and discuss this possibility.	We calculated the model-based loudness as suggested by the reviewer. The results show a small difference, but one that rather suggests a somewhat stronger loudness for the German stimuli. Accordingly, the P1 difference cannot be explained based on loudness. Results of this analysis have been included in lines 193 - 198 in the results section and implication are discussed on lines 370 – 372 in the discussion section “The alternative explanation that a higher perceived loudness of the Chinese stimuli contributes to the effect, which may be due to language-specific differences, was not supported by a computational loudness model.”.
----	--	---

11	"Encoding is based on the complete TRF." You can also perform time-resolved encoding. I highly recommend doing so. This approach can likely shed light on the discrepancy between predictive power and TRF by examining the encoding process over time, which reflects the unfolding of the TRF more accurately and is thus more comparable. For methodological details, you can refer to the work of Hausfeld (2018) or Fiedler (2019).	The encoding analysis has been completely revised. We have also added a time-resolved variant of the encoding to the supplement (Supplementary Fig. 5, 6). The time resolved analysis suggests that the P1 time window dominates the encoding in the complete time window. Moreover, we have added a subset analysis of an exemplary condition, which explains why the Nd effect does not linearly translate into the encoding analysis (Supplementary Fig. 7).
----	--	--

reviewer #2

The manuscript reports on a study in which MEG was used to investigate attention-related modulations of auditory cortex activity during selective listening to native and foreign speech. To this end, groups of Chinese and German native speakers listened to Chinese and German sentences with or without competing input, and in consistent of inconsistent language combinations (i.e., C-C, G-G, G-C, C-G), and detected target words. Noteworthy, all sentences were uttered by the same (bilingual) speaker, thus minimizing the role of acoustic cues for stream segregation.

The data were analyzed using a temporal response function (TRF) approach. For single-speaker speech, differences in P1, N1, and P2 amplitudes between native vs. foreign speech are reported. When listening to competing input, an Nd wave as index of attention

allocation was observed. Nd amplitude was enhanced for native speech, in particular when presented in inconsistent language combinations. The authors further investigated effects of priming on Nd wave amplitude, and compare the results of their TRF approach with a measure of encoding accuracy.

While attentional modulations in selective listening per se have been thoroughly investigated in previous studies, there is currently only little data comparing effects in native vs. foreign speech. The task design is clever, as it should allow to isolate MEG signal modulations related to “native language”, largely independent of differences in low-level acoustic effects. The applied methods and analysis approaches are sound, and are generally well described.

My main concern with this manuscript is the robustness and interpretability of the shown results. The data analysis is based on relatively few trials, and the reported effects may be modulated by participant behavior/perception. As trials cannot be sorted according to participant response (due to the small number of trials), this confound is difficult to address in the statistical analysis. However, it should definitely be considered for the interpretation of the findings. Please find my detailed remarks below:

	Questions	reply
1	The TRF analysis of the two-speaker conditions are based on 30 trials each (i.e, 210 secs of data in total). Previous studies often used training sets > 10 minutes per condition, presumably resulting in a more robust TRF estimation (for a review for effects of data quantity on TRF estimation, see Mesik & Wojczak, 2022 (https://doi.org/10.3389/fnins.2022.963629)). Also, individual trials were rather short compared to prior studies. These factors may reduce the reliability of the measured peak amplitudes as well as the quality of the MEG signal reconstruction. This potential issue should be considered when discussing the current findings. In addition, I suggest providing measures of reliability (Cronbach’s alpha) or SNR for the TRF responses/peaks.	We have added an evaluation of the TRF based on Cronbach's alpha, which shows values around 0.9 for the TRF source waveforms in the Nd time windows (line 215 - 217 “An analysis of data reliability in the Nd time interval, based on Cronbach's alpha calculated across participants and all 16 conditions, revealed a satisfactory values of 0.953 for Chinese 0.918 for German listeners.”). The encoding analysis was changed to a leave-one-out process to enhance the amount of data evaluated. Finally, we discuss what implications the relatively short time intervals per condition may have, in particular for the encoding analysis (TRF versus MEG encoding; line 448 - 474).

2	As depicted in Figure 1, participants showed considerably false alarm rates across all listening conditions, reaching more than 50% in several participants. This suggests frequent failures in auditory stream segregation. As attention is often thought to act on segregated streams and not the acoustic mixture (cf. Shinn-Cunningham 2008), differences between target and masker streams may be larger than reported in the manuscript. Moreover, false alarm rates differ across conditions, so that small attention-related modulations in consistent-language trials may be attributed to failures in auditory stream segregation. To overcome this potential confound, attention-related modulations could be compared within error-free trials only. But, given the small number of trials per condition this seems unfeasible. Therefore, the role of stream segregation for the occurrence of attentional modulations of the TRF should be considered when interpreting the results.	We agree with R2 that the performance in the consistent-language trials - with d' not significantly different from zero - suggests that the two streams cannot be segregated or alternatively are confused as the stimulus proceeds. In this case, even limiting the analysis to error free trials (which are indeed too few for a subanalysis) may produce the same result, because hit and miss trials are overall at chance level, and a lack of a FA could as well be a good guess. To acknowledge these limitations and discuss different options for the interpretation of the present findings, we have added a section in the discussion of stream segregation in the context of the priming paradigm (line 419 - 447).
---	--	---

3	The introduction provides an overview about different factors influencing attention modulations in in two-speaker paradigms. Overall, I felt that it was not optimally tailored towards the actual study. Given the scope of the manuscript on native vs. foreign language processing, I would have expected the authors to provide some literature overview about the listening advantage for native speech, and how this may affect attentional modulations. On the other hand, passages on attention switching and speaker gender, and acoustic degradation do not seem to be overly relevant in the context of the manuscript and could be removed from the introduction. Please also refer to the studies mentioned in comment #6, as these may provide additional context on native vs foreign speech processing.	A limitation is that most of the literature on non-native speech is on L2 speech, i.e. speech that participants can interpret. While we think that our data are also relevant for L2-speech perception under masking, many of the L2 results cannot be used to make predictions for our data, because participants in our study did not understand the speech material. We have tried to make the difference explicit by using (and defining) the term “foreign” speech in the introduction (second paragraph, line 72 - 74 “The comparison of native with foreign speech (i.e. speech in a language that is unknown to the participants), both masked by noise, revealed a trend for higher neural activity and better reconstruction of MEG signals for the foreign speech.”). In response to the reviewer, we have nevertheless extended the part on non-native speech perception under masking and removed the sentence on attention switching instead. Further reference to the L2 literature recommended by the reviewer is taken up in the discussion. We have kept the part on degraded speech, however, because it is similar to foreign speech in so far as it cannot (in the extreme form) be understood. There is an ongoing discussion whether the understandability of degraded speech has an influence on the auditory cortex response related to the envelope. Therefore, predictions based on degraded speech are relevant for the introduction of the experiment, in our view.
---	---	---

4	The TRF analysis is accompanied by an encoding analysis, for better comparability with prior studies. However, if I recall the literature correctly, most of the mentioned studies used multivariate stimulus decoding approaches in which a stimulus feature (mostly, the envelope) is reconstructed from the measured MEG/EEG signals. I am not sure that the results of both analysis types can be directly compared. In broader context, I also wasn't too sure about the relevance of this analysis for understanding the native speech effect. In my view, the rationale of this second analysis needs to be explained more thoroughly, given the room it takes up in the manuscript.	The encoding was not part of our primary hypothesis and we agree with the reviewer that this analysis is not relevant for our conclusions in the end. We initially expected that this analysis would confirm our primary results and thought it would be good to include, as many recent papers report encoding or decoding rather than TRFs. The reviewer is also correct that the studies we directly refer to used decoding rather than encoding, but in our case the two techniques provided very consistent results. Considering comments of reviewers 1 & 3, we have substantially revised the encoding analysis. After all, we think that we can explain relatively well why the TRF and encoding analysis are not linearly related (which we think is of general interest for future comparison between these analysis approaches). In response to Reviewer 2's comment, we have substantially reduced the space used in the main manuscript for reporting the encoding analysis. Most of the additional, exploratory analyses have instead been moved to the supplement. Moreover, we have omitted the parallel report of the decoding analysis, because it appeared redundant and would have taken even more space (note that the similarity of the analysis holds for the new analysis in the revised paper, but was omitted to simplify the presentation).
---	---	---

5	How exactly was the time window for the Nd wave analysis determined? Based on the grand average source waveforms for target and masker streams? For the Chinese speakers, the negative deflection seems to persist beyond this time window, up to 350 – 400 msec. By contrast, the Nd wave seems to be more confined in the German speakers. Have you considered a peak-centered analysis instead of a fixed time window for the Nd wave to account for this group difference?	The Nd time window was defined based on the grand-average TRF of all target trials, starting after the P1 and including the rise of the negative-going response including the first superimposed peak in the grand average after 210 ms. A detailed description has been added in the result section (line 212 – 214 “For the analysis of attention effects, the Nd was defined based on the grand average across all target stream conditions (Supplemental Fig. 4).”), and the grand average used for the definition has been added in Supplementary Fig. 4 and details in the methods section (line 640 “A 100-ms-long time interval from 150 - 250 ms was therefore chosen, starting after the P1 and covering the rising part of the negativity including the first superimposed peak in the grand average.”). We have decided to use a fixed analysis window instead of an individual, peak-based analysis, because the latter is problematic for several reasons: first, not all conditions exhibit a clear N1/Nd response, such that using a peak analysis sometimes measures arbitrary peaks within the noise or returns measures at the end of the time window. Second, the comparison of attended target and unattended masker conditions would typically result in different latencies, which makes a comparison of the two difficult. Generally, the attentional modulation in auditory cortex, i.e. N1 or Nd, has been described as transient after P1, which is why we did not extend the time interval further to also cover the full sustained response. We agree with the reviewer that the separation of where the transient interval ends and the sustained interval starts is difficult due to ambiguous indicators for boundaries.
---	--	--

6	Multiple previous studies have compared speech-related activity in native vs foreign speech, partially also using natural speech and similar TRF or encoding/decoding approach (e.g., Ihara et al, 2021 (https://doi.org/10.3389/fnhum.2021.665809), Brodbeck et al., 2023(https://doi.org/10.1523/JNEUROSCI.0666-23.2023), Di Liberto et al., 2021 (https://doi.org/10.1016/j.neuroimage.2020.117586), Zinsner et al., 2022 (https://doi.org/10.1016/j.bandl.2022.105128)). These studies seem relevant for the current experiment, as they provide further insights into the processing of native vs foreign speech, which may help explaining the observed native speech effects in the single-speaker condition.	We have added three of the studies to the discussion (line 486, line 491, and line 496). We found it difficult to relate the study by Ihara et al to our data, first, because the analysis is mainly on speech proficiency and, second, because the analysis is based on word onset, both of which cannot be directly compared to our data. Note however that these studies are also on L1 versus L2 and not on L1 versus foreign speech (cf. reply to comment 3).
7	As I understand the reported experimental procedures, the non-primed listening trials were presented first, followed by the primed trials. Was this counterbalanced across subjects? If not, could some of the observed behavioral differences be explained by training or time on task?	This is correct, the primed stimuli were always presented in the second half of the recording session. Thus, we cannot exclude any training effects that could additionally contribute to the difference between primed and unprimed conditions. On the other hand, since a general difference in Nd amplitude was not observed but only an enhanced negative-going response for target and masker streams alike, we consider the possible influence of practice less likely. We added a sentence in the results section (line 272 “Thus, an additional effect of practice in addition to priming cannot be excluded.”) and one in the discussion (line 446 “Finally, since unprimed blocks always preceded primed blocks in the current

		study, a practice effect can also not be ruled out completely.”) to point out the possibility of practice effects.
8	Across the manuscript, d' is reported as “stronger”. If I am not mistaken, It should read “higher”.	Thank you for spotting, this has been corrected.
9	The description “little experience of German“ for the Chinese speakers seems vague. What was the inclusion criterion (in terms of length of residency, German class participation, etc.)? Please clarify.	Most of the participants had done a four-weeks German class before moving to Heidelberg, but they did not report significant understanding of German. This and the duration that participants had been living in Germany has been added to the methods section (line 516 – 519 “13 of the Chinese participants had participated in a one-month German course before moving to Germany, but none of them reported to understand German when they participated in the experiment. The average time that Chinese participants had spent in Germany was 8.5 months (3 - 15months).”).
10	In the primed trials, were the target streams presented directly before the mixture or were there some trials in between? Please clarify the corresponding section of the methods.	Primes were presented directly before the corresponding mixtures. This information has been added in the beginning of the results section (line 107 “In the second half of the experiment, the primed condition was presented, where the complete target stream was presented alone directly before the presentation of the mixture.”) and in the methods

		section (line 545 “In the second section of the experiment, the target streams were presented once as prime before the mixtures.”).
11	It seems that the lambda parameter was optimized using a grid search and LOOCV. Please specify the tested lambda range in the manuscript. Also, for comparison with others studies please provide the resulting lambda parameter that was used for data analysis.	The range for search and the resulting lambda values have been added in the methods section (line 594 – 596 “The optimal lambda parameter was searched for in the range from 10^{-6} to 10^{10} , yielding values in a range from 10^3 to 10^6 (6×10^3 , 27×10^5 , 1×10^6).”).
12	Given that the TRF was computed on the sensor level, I suggest providing a (supplementary) figure showing the TRF topography in sensor space as well as information about the dipole fit (x,y,z coordinates, direction etc.).	A figure that maps the TRF topography for prime, target, and masker TRFs has been added as Supplementary Fig. 3. We also added the dipole coordinates in the new Supplementary Table 1 and also show them in the new Figure panel e of Fig. 2 in the main manuscript.
13	Differences in envelope peak rate are linked to differences in syllable rate in the discussion. This assumption should be checked for the used stimulus material (e.g., using an online tool such as webmaus for German).	the syllable rate has been measured and added to the manuscript for comparison (results, line 190 – 191 “The syllable rate amounted to 5.4/s in German and 4.2 in Chinese, confirming that envelope peak rate and syllable rate are proportional to each other.”).

14	Thank you for providing the behavioral and MEG data of the experiment in an online data repository. However, I would like to point out that the reported findings can only be reproduced by others if the stimulus material or the used envelope regressors are also shared with the neural data. I therefore suggest including these data as well.	The data have now been uploaded to a repository including the regressors required for their evaluation under a private URL: https://heidata.uni-heidelberg.de/privateurl.xhtml?token=0ad0078f-3e32-470d-8b2b-a357b5eb6df3
----	---	--

reviewer #3

Liang, Gerwien and Gutschalk present data and results of an MEG study on listening to audiobook narrations consisting of native and/or non-native language. Participants (native Chinese and German speakers) were presented with short auditory scenes of 7s of either a single speech stream (SS) or two streams (TS) of the same speaker and were asked to press a button when they heard the target word for the relevant speaker. The authors varied target-masker language consistency (same or different language), target language (Chinese or German), priming (primed nor not primed) and acquired MEG data. Behavioral results indicated effects of each of these factors including their interactions. The authors performed an encoding analysis looking at the TRFs for source-localized activity based on the observed P1 in SS and focused on the coefficients for the delay 150-250ms for TS. The main findings were differences for target or masker stimuli, target-masker consistency, priming and an interaction between target language and participant group. Looking at the TRF model performances, the results were different to the analysis of the TRF coefficients. The authors conclude that attention enhanced the target speech more 1) when it was in the native vs non-native language for both SS and TS and 2) when the masker stream was in a different language.

While the study has great appeal and is of clear interest to the community, in particular regarding the bilingual aspect (same narrator for both languages, large set of participants in each of the native groups), and the figures are very transparent by including single-participant points, methodological and design aspects need to be addressed before judging the results. Please find my main and minor points below.

	Questions	reply
1	TRF analysis i. The authors main results are based on the estimated TRFs that rely on an encoding analysis (via mTRFs/ridge regression). Compared to earlier studies, the stimuli are rather short (7s) which makes it difficult to get a reliable estimate for single trials. The task choice of target detection makes this even more difficult as targets (their detection and the button press) will undoubtedly contribute to the MEG signal and thus TRF estimation. To address this, one option that would not require collecting new data would be to include event predictors for the target onset in the target and masker speech as well as button press onsets (at least 3 predictors; a differentiation could be made for detected vs missed targets leading then to 5 ‘confound’ predictors). As it is an encoding analysis, these can be included in the very same estimation in addition to the envelope derivative; the resulting TRF for the envelope derivative could then be used for further analysis (see e.g. Orb et al., 2023, iScience). This would result in responses closer to the relevant auditory activity as the other predictors would capture (most, not all) variance explained by processes unrelated to the envelope tracking. Looking at TRFs of the other predictors could then provide an ERP-like response to the coded events. As is, the TRFs and their analyses are difficult to interpret.	The data have been re-analyzed as suggested (i.e. using the envelope from the previous analysis, the onset of the target word, the onset of the same word in the distractor, and the response as fifths event), modeling target, distractor "Jack", and responses as additional events. The results are presented in the revised paper instead of the previous analysis.

2	TRF analysis ii. Similar to the point above, by using the encoding framework, one main advantage is that TRFs for different predictors can be estimated at once. This holds also for target and masker envelopes and including these to estimate the TRFs would remove/reduce the shared components and better delineate the TRFs for target and masker envelope derivatives.	The target and masker envelope had already been estimated together in the initial version of the analysis, this was maybe not explained well enough, the methods description of the encoding analysis has been revised. (Note that we did not include the additional regressors for the encoding analysis, because the latter is now focused on the auditory cortex and the additional components are mostly generated elsewhere).
3	Priming. The motivation behind the priming manipulation remains unclear to me (I agree that it had a strong effect by making the task clearly less difficult as the same segment was repeated) and by including the interpretation of the data becomes arguably more complex. To me, the prime trials render the listening less natural as it already is (target detection of a word will be different than listening to speech; however, it constitutes a task that can be performed in known and unknown language) and I would consider the unprimed data to better reflect natural listening. Please include the motivation.	The priming task was motivated for comparability with the (as far as we know) only MEG/EEG study that had used the same speaker in a mixture before (Wang et al. 2019). Since that study had suggested that an attention effect was only observed with priming, we had included this condition to avoid seeing no effects at all. This information has been added in the motivation at the end of the introduction (line 97 “As a previous study with a same-speaker setup had found an attention effect only when the target stream was primed with the isolated target stream, we added a comparable condition to our setup.”).

4	The Nd hypothesis (l.87/88) is not integrated in the introduction and, as presented, might appear as being included after data inspection. Please introduce the Nd better or keep it more general by saying that you expect differences in the TRF. Similarly, the specification of the intervals after looking at the data requires careful analysis to avoid double-dipping. Please report which data and the statistics that was considered to define the intervals (for SS and, more importantly, TS) for readers to better judge the selection.	We now introduce the Nd concept in the revised introduction (line 60 “In the context of speech streams, this negative-going wave has mostly been referred to as N1/N100 whereas studies investigating attention to tone streams have variably referred to it as N1, processing negativity, or neutrally to the negative difference wave (Nd)”). The observation that attentional modulation in auditory cortex is mostly observed by negative-going responses is quite accepted in the literature using tone streams, whereas the nomenclature has been somewhat different for the attention to speech streams. Another motivation for using Nd rather than N1 here is our own work, where we previously found negative-going activity for attended streams (referred to N1 by that time) that overlapped in latency with the P2, the latter of which was mostly observed in unattended trials, only. The time intervals for the statistical analysis were defined based on the grand average for the primes and the grand average for the target streams, such that their definition should have no influence on the comparisons that are of interest for this study (in the case of Nd it could be argued that the overall attention effect was biased by this choice. However, this does not apply to the interactions with the attention effect, which are the research question of this study). The procedure is now described in more detail in the methods section (line 640 “A 100-ms-long time interval from 150 - 250 ms was therefore chosen, starting after the P1 and covering the rising part of the negativity including the first superimposed peak in the grand average.”), referenced in the
---	---	---

		results section (line 212 – 214 “For the analysis of attention effects, the Nd time window was defined based on the grand average across all target stream conditions (Supplemental Fig. 4).”), and shown in the new supplementary Fig. 4. During the literature review for this revision, we realized that a number of papers also report enhanced P2 for attended speech streams, but did not find any discussion/explanation of/for why some studies find this and other (like ours) do not. We have therefore referenced this point in the discussion (in the context of the encoding analysis, for which it is particularly relevant; see line 450 - 460).
--	--	---

5	Methodological detail (TRF approach). a) I did not find information whether the first second of the stimuli was cut for the analysis. It would be more elegant to do so, as the TRF of TS will otherwise include speech without a masker material. One could argue that the analysis shouldn't start until about .5-1s after the onset of the masker stimulus as the onset of speech will affect the TRF strongly. b) It is unclear whether the authors performed permutations for the encoding/TRF. These can be used for both the TRF analysis as well as the 'encoding analysis' by providing an empirical chance level for the TRF coefficients as well as the encoding accuracy (one could argue that this is not necessary as the the comparison of TRF/encoding performance are between conditions). c) It is unclear to me whether the TRF estimation was performed on different data (P1 sources vs sensors for each hemisphere). I think it is the latter (l.297/8) and cannot come up with a reason why the analysis was redone, on different signals. An encoding analysis with mTRFs will provide both the estimated TRFs and a performance measure on how well this TRF can predict unseen data. This would make it more comparable and might already reduce the differences observed between TRF and encoding performance already. d) please provide the range of lambdas to optimize the models.	(a) the first second was cut from the streams, more details have been added in methods (line 579 "For the analysis of the mixtures, the cues - i.e. the first second of the target stimuli - were not included in the analysis."). In the process of this revision, we have also tested a version where further 0.5 s at the beginning of the second stream were omitted. While this analysis produced generally the same results in the statistical analysis in the Nd time window (the effects were even somewhat stronger), the waveforms coming out of this analysis were more noisy, probably because of the further reduction of stimulus material. We therefore decided to stick with the time intervals that include the onset of the masker stream. (b) We did not perform a permutation test. In our understanding this is not required for the comparison performed here. (c) We have now found a way to transform the raw data into source waveforms and have changed the encoding analysis to be in source space, as well. The results are rather similar to the previous, sensor-based analysis, and did not resolve the differences between the TRF and encoding results. (d) Lambda values have been added to the methods section (line 594 – 596 "The optimal lambda parameter was searched for in the range from 10⁻⁶ to 10¹⁰, yielding values in a range from 10³ to 10⁶ (6x 10³, 27x 10⁵, 1x10⁶).").
---	---	---

6	It is unclear why reaction times were not analysed. These can provide a more sensitive, continuous outcome compared to accuracy scores with regard to task difficulty.	Reaction times have now been evaluated and produced overall similar results as the d' analysis (except for the comparison between primed and unprimed trials). These results are now briefly mentioned in the results section and are shown in supplementary Fig. 1 and 2.
7	Often the text about TRF results discusses '(un)attended conditions/trials', which were confusing as there are no attention conditions (participants always pay attention). Rephrasing this as 'target(masker) speech/streams' or '(un)attended speech/streams' could make this easier to follow.	The text has been edited as suggested.
8	Please mention that there were no effects of hemisphere already in the beginning of the TRF analysis section (starting at l.136) to explain that the presented results are pooled across hemispheres.	This information has been added in line 164 - 166("Since no hemisphere main effects or interactions with hemisphere were observed, the average of left- and right hemisphere activity is reported below.") and in line 217 – 220("For the attention effect in the Nd time interval, no significant main effects of or first-order interactions with hemisphere were observed, such that the statistics below do not report any hemisphere effects for clarity.").
9	For better clarity, please provide effect size measures (for example in the ANOVA tables) and mean+/-sem for descriptive values (e.g. d-prime or TRF coefficients). For example, adding it to Table S1 and provide the same for the TS condition.	Effect-size values (partial eta square) have been added to all tables. Mean and sem are provided in the figures for the single conditions.

10	The data set is very interesting to the community. However, I'm missing a data availability statement.	The data are available online for review and will be published in final version together with the paper.
----	--	--

a Chinese Native Listeners

c

e

b German Native Listeners

d

f

updates:

1. reponses time window changed from 2s time window from onset of the target word to offset of the target word
2. Only the button presses that appeared in the 2s time window after offset of the target word from unattended streams were considered as false alarms.

updates:

1. Average dipole positions fitted to the P1 projected on a standard brain were added.
2. Hit rate for speech in quiet was included in this figure.
3. In addition to envelope, two more predictors (target word position and button presses) were given to the regression.

updates:

1. Five predictors (attended speech envelope, unattended speech envelope, target word from attended speech, target word from unattended speech and button presses) was included.

updates:

1. Five predictors (attended speech envelope, unattended speech envelope, target word from attended speech, target word from unattended speech and button presses) was included.

updates:

1. The prediction accuracy shown here was the mean from leave-one-out cross validation(all data) instead of prediction accuracy of unseen data.
2. The lambda was the optimal lambda fitted to envelope of speech in quiet.